# A de novo paradigm for male infertility

M. S. Oud [1,42], R. M. Smits[2,42], H. E. Smith[3,42], F. K. Mastrorosa [3], G. S. Holt[3], B. J. Houston[4], P. F. de Vries[1],
B. K. S. Alobaidi [3], L. E. Batty[3], H. Ismail[3], J. Greenwood[5], H. Sheth [6], A. Mikulasova[3], G. D. N. Astuti[7,8],
C. Gilissen [7], K. McEleny[9], H. Turner[10], J. Coxhead [11], S. Cockell[12], D. D. M. Braat [2], K. Fleischer[2],
K. W. M. D'Hauwers[13], E. Schaafsma[14], Genetics of Male Infertility Initiative (GEMINI) consortium*,
L. Nagirnaja [15], D. F. Conrad [15], C. Friedrich[16], S. Kliesch [17], K. I. Aston[18], A. Riera-Escamilla[19], C. Krausz[20],
C. Gonzaga-Jauregui[21], M. Santibanez-Koref[3], D. J. Elliott[3], L. E. L. M. Vissers [1], F. Tüttelmann [16],
M. K. O'Bryan [4], L. Ramos [2], M. J. Xavier [3,43], G. W. van der Heijden [1,2,43] & J. A. Veltman [3,43 ✉]

De novo mutations are known to play a prominent role in sporadic disorders with reduced fitness. We hypothesize that de novo mutations play an important role in severe male infertility and explain a portion of the genetic causes of this understudied disorder. To test this hypothesis, we utilize trio-based exome sequencing in a cohort of 185 infertile males and their unaffected parents. Following a systematic analysis, 29 of 145 rare (MAF < 0.1%) protein-altering de novo mutations are classified as possibly causative of the male infertility phenotype. We observed a significant enrichment of loss-of-function de novo mutations in loss-of-function-intolerant genes ($p$-value = $1.00 \times 10^{-5}$) in infertile men compared to controls. Additionally, we detected a significant increase in predicted pathogenic de novo missense mutations affecting missense-intolerant genes ($p$-value = $5.01 \times 10^{-4}$) in contrast to predicted benign de novo mutations. One gene we identify, *RBM5*, is an essential regulator of male germ cell pre-mRNA splicing and has been previously implicated in male infertility in mice. In a follow-up study, 6 rare pathogenic missense mutations affecting this gene are observed in a cohort of 2,506 infertile patients, whilst we find no such mutations in a cohort of 5,784 fertile men ($p$-value = 0.03). Our results provide evidence for the role of de novo mutations in severe male infertility and point to new candidate genes affecting fertility.

A full list of author affiliations appears at the end of the paper.

Male infertility contributes to approximately half of all cases of infertility and affects 7% of the male population. For the majority of these men the cause remains unexplained[1]. Despite a clear role for genetic causes in male infertility, there is a distinct lack of diagnostically relevant genes and at least 40% of all cases are classified as idiopathic[1–4]. Previous studies in other conditions with reproductive lethality, such as neurodevelopmental disorders, have demonstrated an important role for de novo mutations (DNMs) in their etiology[5]. In line with this, recurrent de novo chromosomal abnormalities play an important role in male infertility. Both azoospermia factor (AZF) deletions on the Y chromosome as well as an additional X chromosome, resulting in Klinefelter syndrome, occur de novo. Collectively, these de novo events explain up to 25% of all cases of nonobstructive azoospermia (NOA)[1,4]. Interestingly, in 1999 a DNM in the Y-chromosomal gene *USP9Y* was reported in a man with azoospermia[6]. Until now, however, a systematic analysis of the role of DNMs in male infertility had not been attempted, even though a pilot exome sequencing study in 13 infertile men and their parents was recently published[7]. This is partly explained by a lack of basic research in male reproductive health in general[4,8], but also by the practical challenges of collecting parental samples for this disorder, which is typically diagnosed in adults.

In this work, we address this lack of knowledge by analysing exome sequencing data of 185 infertile males and their parents and reporting on our findings of 29 DNM in these men which are likely causative for the infertility phenotype, based on variant and gene level evidence. We emphasize an enrichment for loss-of-function (LoF) DNM in LoF-intolerant genes and missense DNM in missense-intolerant genes. We identify a number of promising candidate genes for male infertility, including the mRNA splicing gene *RBM5*, which contains a possibly causative DNM in our trio cohort, and possibly causative heterozygous variants in six additional patients for which parental information is not available. This work suggests a potential role for DNM as a cause of severe male infertility and addresses the need for further investigation in larger patient–parent trio cohorts to solidify these results.

## Results

**Discovery of de novo mutations in infertile male trios**. In this study, we investigated the role of DNMs in 185 unexplained cases of oligozoospermia (<5 million sperm cells/ml; $n = 74$) and azoospermia ($n = 111$) by performing whole exome sequencing (WES) in all patients and their parents (see Supplementary Figs. 1, 2, Supplementary Notes and Data for details on methodology and clinical descriptions). In total, we identified and validated 192 rare DNMs (MAF < 0.1%), including 145 protein-altering DNMs. All de novo point mutations were autosomal, except for one on chromosome X, and all occurred in different genes (Supplementary Data 1). Two rare de novo copy-number variations (CNVs) were also identified affecting a total of 7 genes (Supplementary Fig. 3). None of the 145 protein-altering DNMs occurred in a gene already known for its involvement in autosomal dominant human male infertility. This is not unexpected as only four autosomal dominant genes have so far been linked to isolated male infertility in humans[3,9].

**Intolerance analysis of genes with de novo loss-of-function mutations**. Broadly speaking, across genetic disorders, dominantly acting disease genes are usually intolerant to LoF mutations, as represented by a high pLI score[10] or a low LOEUF score[11]. In our cohort of infertile men, we detected a significant enrichment in the number of LoF-intolerant genes with a LoF DNM ($n = 17$). No such enrichment was identified in a cohort of 1,941 control cases from de novo-db v1.6.1[12] (median pLI in patients with male infertility = 0.80, median pLI in controls = $3.75 \times 10^{-5}$, p value = $1.00 \times 10^{-5}$, N simulations = 100,000) (Fig. 1a). Similar results were obtained using the LOEUF scores

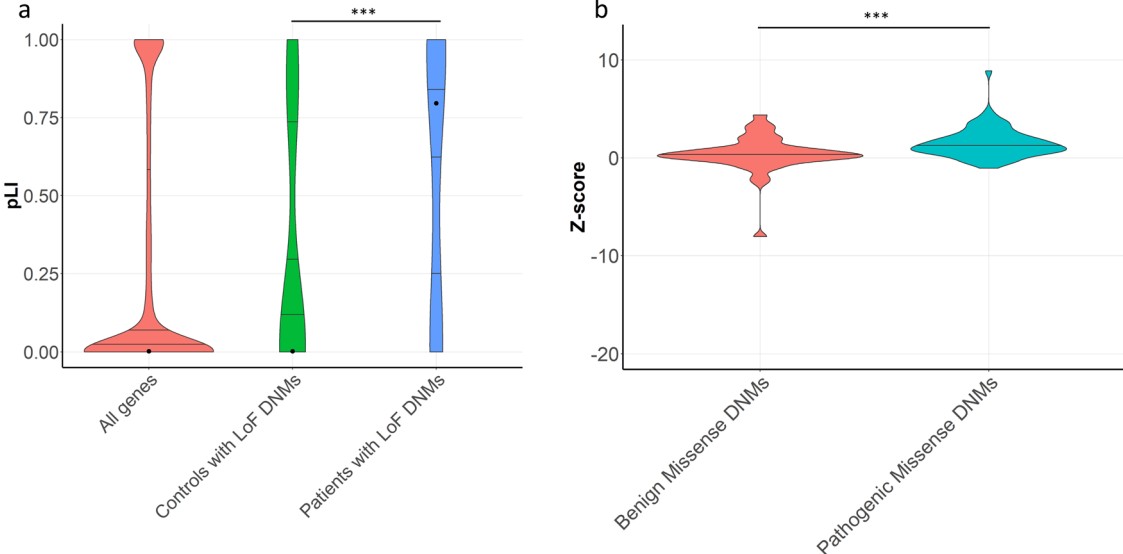

**Fig. 1 Analysis of the intolerance to loss-of-function and missense variation in genes with de novo mutations. a** Violin plot with quantile lines showing pLI scores in all genes in gnomAD (red), all genes affected by rare protein-altering loss-of-function (LoF) de novo mutations (DNMs) in a control population (http://de novo-db.gs.washington.edu/de novo-db/) (green) and in all genes with a rare protein-altering LoF DNM in our trio cohort (blue). Using the permutation-based, nonparametric test defined by Lelieveld et al. [64] a significant enrichment of LoF DNMs in LoF-intolerant genes in patient cohort was detected in comparison to the number of LoF in fertile control cohort (DNM LoF mutations in patients $n = 17$, median pLI in patients with male infertility = 0.80, DNM LoF mutations in controls $n = 21$, median pLI in controls = $3.75 \times 10^{-5}$, p value = $1.00 \times 10^{-5}$, N simulations = 100,000). The black dot indicates median pLI scores. **b** Violin plot with quantile lines showing the distribution of Z-scores for genes with predicted benign ($n = 59$) and pathogenic missense DNMs ($n = 63$) in infertile patients. A significant increase in predicated pathogenic DNMs in missense-intolerant genes was detected compared to benign missense DNM (Two-sided Mann–Whitney U test, p value of $3.44 \times 10^{-4}$). (***p value < 0.001).

(median LOEUF in patients with male infertility = 0.34, median LOEUF in controls = 0.59, $p$ value = $1.00 \times 10^{-5}$, N simulations = 100,000) (Supplementary Fig. 4a). This observation indicates that LoF DNMs likely play an important role in male infertility, similar to what is known for developmental disorders and severe intellectual disability[13,14]. As an example, a heterozygous likely pathogenic frameshift DNM was observed in the LoF-intolerant gene *GREB1L* (pLI = 1) of Proband_076. Homozygous *Greb1l* knockout mice appear to be embryonic lethal, however, typical male infertility phenotypic features such as abnormal fetal testis morphology and decreased fetal testis volume are observed[15]. Interestingly, this patient has a reduced testis volume and severe oligozoospermia (Supplementary Notes Table 1). Nonsense and missense mutations in *GREB1L* in humans are known to cause renal agenesis[16] (OMIM: 617805), not known to be present in our patient. Of note, all previously reported damaging mutations in *GREB1L* causing renal agenesis are either maternally inherited or occurred de novo. This led the authors of one of these renal agenesis studies to speculate that disruption to *GREB1L* could cause infertility in males[15]. A recent WES study involving a cohort of 285 infertile men also noted several patients presenting with pathogenic mutations in genes with an associated systemic disease where male fertility is not always assessed[17].

We also assessed the damaging effects of the two rare de novo CNVs by looking at the pLI score of the genes involved. Proband_066 presented with a large 656 kb de novo deletion on chromosome 11, spanning 6 genes in total. This deletion partially overlapped with a deletion reported in 2014 in a patient with cryptorchidism and NOA[18]. Two genes affected in both patients, *QSER1* and *CSTF3*, are LoF-intolerant with pLI scores of 1 and 0.98, respectively. In particular, *CSTF3* is highly expressed within the testis and is known to be involved in pre-mRNA 3′-end cleavage and polyadenylation[19].

**Missense intolerance in de novo mutation genes**. To systematically evaluate and predict the likelihood of these DNMs causing male infertility and identify novel candidate disease genes, we assessed the predicted pathogenicity of all DNMs using three prediction methods based on SIFT[20], MutationTaster[21], and PolyPhen2[22] with a minimum of 2 of the 3 showing pathogenicity to define a variant as Pathogenic. Using this approach, 84 of 145 rare protein-altering DNM were predicted to be pathogenic, while the remaining 61 were predicted to be benign. To further analyse the impact of the variants on the genes affected, we looked at the missense Z-score of all 122 genes affected by a missense variant, which indicates the tolerance of genes to missense mutations[23]. We identified no significant enrichment in missense DNMs in missense-intolerant genes in our infertile cohort when compared to controls (median Z-score in male infertility patients = 0.83, median Z-score in controls = 1.04, $p$ value = 1, N simulations = 100,000) (Supplementary Fig. 4b). Interestingly, however, we observed a significantly higher median missense Z-score in genes affected by a missense DNM predicted as pathogenic (median Z-score = 1.21, $n$ = 63) when compared to genes affected by predicted benign (median Z-score = 0.98, $n$ = 59) missense DNMs in our cohort ($p$ value = $5.01 \times 10^{-4}$, Fig. 1b). It should be noted that the same analysis in controls showed no such significant difference (Supplementary Fig. 4c).

**Protein–protein interactions reveal link to mRNA splicing**. An analysis using the STRING database[24], revealed a significant enrichment of protein interactions amongst the 84 genes affected by a protein-altering DNM predicted to be pathogenic (PPI enrichment $p$ value = $2.35 \times 10^{-2}$, Fig. 2). No such enrichment

was observed for the genes highlighted as likely benign ($n$ = 61, PPI enrichment $p$ value = 0.206) or those affected by synonymous DNMs ($n$ = 35, PPI enrichment $p$ value = 0.992, Supplementary Fig. 5). This suggests that the proteins affected by predicted pathogenic DNMs share common biological functions.

The STRING network analysis also highlighted a central module of interconnected proteins with a significant enrichment of genes required for mRNA splicing (Supplementary Fig. 6). The genes *U2AF2*, *HNRNPL*, *CDC5L*, *CWC27*, and *RBM5* all contain predicted pathogenic DNMs and likely interact at a protein level during the mRNA splicing process. Pre-mRNA splicing allows gene functions to be expanded by creating alternative splice variants of gene products and is highly elaborated within the testis[25]. One of these genes, *RBM5* has been previously highlighted as an essential regulator of haploid male germ-cell pre-mRNA splicing and male fertility in mice[26]. Mice with a homozygous ENU-induced allele point mutation in *RBM5* present with azoospermia and germ cell development arrest at round spermatids. Whilst in mice, a homozygous mutation in *RBM5* is required to cause azoospermia, this may not be the case in humans as is well-documented for other genes[27], including the recently reported male infertility gene *SYCP2*[9]. Of note, *RBM5* is a tumor suppressor in the lung[28], with reduced expression affecting RNA splicing in patients with non-small cell lung cancer[29]. *HNRNPL* is another splicing factor affected by a possible pathogenic DNM in our study. One study implicated a role for *HNRNPL* in patients with Sertoli cell-only phenotype[30]. The remaining three mRNA splicing genes have not yet been implicated in human male infertility. However, mRNA for all three is expressed at medium to high levels in human germ cells and all are widely expressed during spermatogenesis[31]. Specifically, *CDC5L* is a component of the PRP19-CDC5L complex that forms an integral part of the spliceosome and is required for activating pre-mRNA splicing[32], as is *CWC27*[33]. *U2AF2* plays a role in pre-mRNA splicing and 3′-end processing[34]. Interestingly, *CSTF3*, one of the genes affected by a de novo CNV in Proband_066, affects the same mRNA pathway[18].

**DNMs uncovering recessive disease and analysis of maternally inherited mutations**. Whilst DNMs most often cause dominant disease, they can contribute to recessive disease, usually in combination with an inherited variant on the trans allele. In order to look for this, we analysed all DNM genes for the presence of inherited mutations on the other allele in the same patient. In Proband_060, who carried a DNM in Testis and Ovary Specific PAZ Domain Containing 1 (*TOPAZ1*) on the paternal allele, we did identify a maternally inherited variant predicted to be pathogenic (Supplementary Fig. 7). *TOPAZ1* is a germ cell-specific gene which is highly conserved in vertebrates[35]. Studies in mice revealed that *Topaz1* plays a crucial role in spermatocyte, but not oocyte, progression through meiosis[36]. In men, *TOPAZ1* is expressed in germ cells in both sexes[31,37,38]. Analysis of the testicular biopsy of this patient revealed a germ cell arrest in early spermiogenesis (Fig. 3).

Maternally inherited mutations can also result in dominant causes of male infertility if not affecting female fertility. We therefore studied all DNM genes for the presence of maternally inherited mutations in the entire cohort and compared this to the presence of paternally inherited mutations in the same genes. A total of 4 maternally inherited variants predicted to be pathogenic were identified in DNM genes (*TENM2* (2×), *CWC25*, and *EVC*). All of these variants, however, were also observed multiple times in an exome dataset from a cohort of 5784 fertile men suggesting that these maternally inherited variants are not causative of male infertility (Supplementary Data 2).

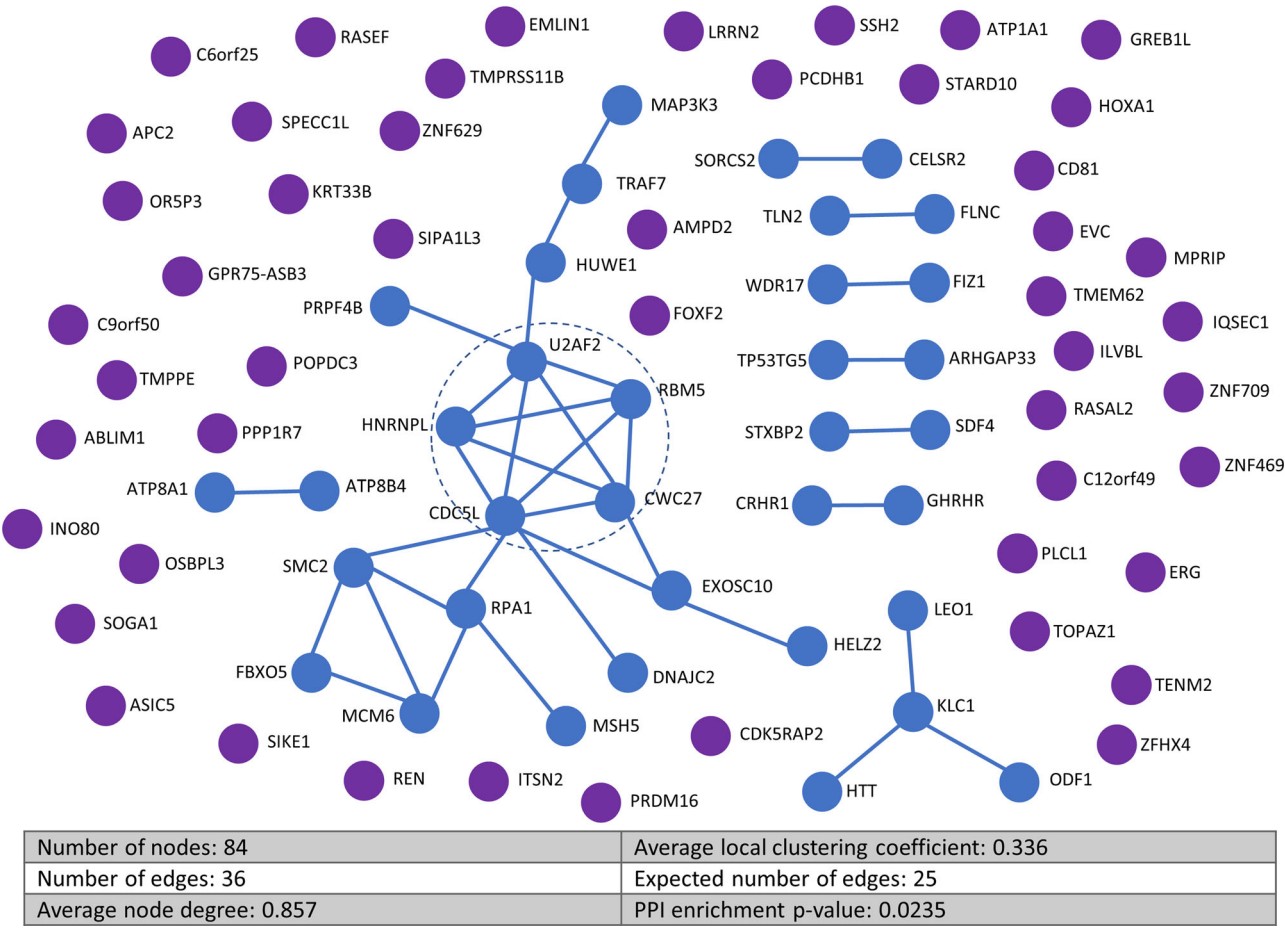

| Number of nodes: 84 | Average local clustering coefficient: 0.336 |
| Number of edges: 36 | Expected number of edges: 25 |
| Average node degree: 0.857 | PPI enrichment p-value: 0.0235 |

**Fig. 2 Protein–protein interactions predicted for proteins affected by pathogenic de novo mutations.** Significantly larger number of interactions were observed in proteins affected by de novo mutations than expected for a similar sized dataset of randomly selected proteins (PPI enrichment $p$ value = $2.35 \times 10^{-2}$). The central module of the main interaction network (blue dashed circle) contains 5 proteins involved in mRNA splicing (Supplementary Fig. 6).

**Further analysis in additional cohorts of infertile males.** In addition to all systematic analyses described above, we evaluated the function of all DNM genes to give each a final pathogenicity classification (Table 1, details in Methods). Of all 192 DNMs, 29 affected genes were linked to male reproduction and classified as possibly causative, with a further 50 as unclear. For replication purposes, only one pilot study including 13 trios was recently published in male infertility[7]. None of the DNM genes reported in this study showed DNMs in our cohort. To further study the DNM genes identified in our cohort, we looked for the presence of rare predicted pathogenic mutations in these genes in exome datasets of infertile men ($n = 2,506$), in collaboration with members of the International Male Infertility Genomics Consortium and the Geisinger-Regeneron DiscovEHR collaboration[39]. For comparison, we included an exome dataset from a cohort of 11,587 fertile men and women from Radboudumc.

In the additional infertile cohorts, we identified 17 LoF mutations in our DNM LoF-intolerant genes (pLI ≥ 0.9), although we did not detect a statistical enrichment in the LoF mutations in these genes compared to fertile men (Two-tailed Fisher's Exact test with Bonferroni correction adjusted $p$ values >0.05, Supplementary Data 3, 4). Next, we looked for an enrichment of rare predicted pathogenic missense mutations in these cohorts (Table 2 and Supplementary Data 5, 6). A total of 11 genes showed an enrichment of pathogenic missense mutations in infertile men compared to fertile men (Two-tailed

Fisher's Exact test, $p$ value < 0.05, Table 2). After applying the Bonferroni correction to counteract the effects of multiple testing, however, the only significant enrichment was observed in the *RBM5* gene (adjusted $p$ value = 0.03). In this gene, six infertile men were found to carry a rare pathogenic missense mutation, in addition to the proband with a de novo missense mutation (Supplementary Fig. 8, Supplementary Data 7). Importantly, no such predicted pathogenic mutations were identified in men in the fertile cohort. In line with these results, *RBM5*, already highlighted above as an essential regulator of male germ cell pre-mRNA splicing and male infertility[26], is highly intolerant to missense mutations (missense Z-score 4.17).

In addition to the comparison between fertile and infertile men, we investigated whether there was any difference between the number of predicted pathogenic mutations carried in fertile men compared to fertile women. However, none of the DNM genes showed a significant difference between the sexes (Two-tailed Fisher's Exact test with Bonferroni correction adjusted $p$ values = 1, Supplementary Data 3, 5).

**Phasing of de novo mutations to identify parent of origin.** Given the predicted impact of these DNMs on spermatogenesis, we were interested in investigating the parental origin of DNMs in our trio cohort. We were able to phase 29% ($n = 59$) of all our DNMs using a combination of short-read WES and targeted long-read sequencing (Supplementary Table 1). In agreement

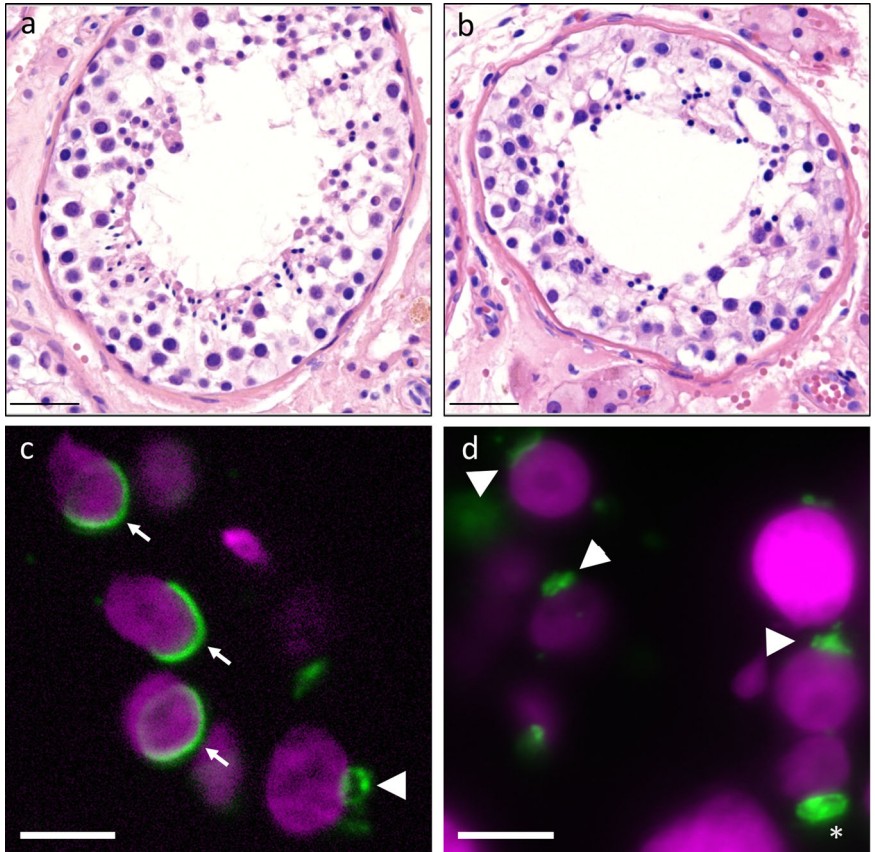

**Fig. 3 Description of control and *TOPAZ1* proband testis histology and aberrant acrosome formation. a**, **b** H&E stainings of (**a**) control and (**b**) Proband_060 with pathogenic mutations in *TOPAZ1* gene. The epithelium of the seminiferous tubules in the *TOPAZ1* proband show reduced numbers of germ cells and an absence of elongating spermatids based on the analysis of 150 seminiferous tubules in control and patient. **c**, **d** immunofluorescent labeling of DNA (magenta) and the acrosome (green) in control sections (**c**) and *TOPAZ1* proband sections (**d**). (**c**) The arrowhead indicates the acrosome in an early round spermatid and the arrows the acrosome in elongating spermatids. Spreading of the acrosome and nuclear elongation are hallmarks of spermatid maturation. (**d**) No acrosomal spreading (see arrowheads) or nuclear elongation is observed in the *TOPAZ1* proband. The asterisk indicates an example of progressive acrosome accumulation without spreading. Scale bar: 40 μm (**a**, **b**) and 5 μm (**c**, **d**).

**Table 1 De novo mutation classification summary.**

|  | Possibly causative | Unclear | Unlikely causative | Not causative | Total |
|---|---|---|---|---|---|
| Missense | 21 | 38 | 50 | 13 | 122 |
| Frameshift | 4 | 8 | 1 | 0 | 13 |
| Stop gained | 1 | 3 | 0 | 0 | 4 |
| In-frame indels | 3 | 1 | 1 | 1 | 6 |
| Splice site variant | 0 | 0 | 0 | 11 | 11 |
| Synonymous | 0 | 0 | 0 | 36 | 36 |
| TOTAL | 29 | 50 | 52 | 61 | 192 |

Rare DNMs were classified based on pathogenicity prediction, ACMG classification, number of cases in gnomAD and presence of the exact mutation in verified fertile men of the control cohort, as well as functional data taking into account RNA expression in testis, RNA enrichment in the testis or involvement in spermatogenesis, protein expression in the testis, model organisms, the protein function in relation to spermatogenesis and interactions with known fertility genes.

with literature[40–43], 72% of all DNMs occurred on the paternal allele. Interestingly, phasing of 8 likely causative DNMs showed that 6 of these were of paternal origin (75%). This suggests that DNMs with a deleterious effect on the future germline can escape negative selection in the paternal germline. This may be possible because the DNM occurred after the developmental window in which the gene is active, or the DNM may have affected a gene in the gamete's genome that is critical for somatic cells supporting the (future) germline. Transmission of pathogenic DNMs may

also be facilitated by the fact that from spermatogonia onwards, male germ cells form cysts and share mRNAs and proteins[44]. As such, the interconnectedness of male germ cells, which is essential for their survival[45], could mask detrimental effects of DNMs occurring during spermatogenesis.

## Discussion

In 2010, we published a pilot study pointing to a de novo paradigm for mental retardation[46] (now more appropriately termed developmental disorders or intellectual disability). This work contributed to the widespread implementation of patient–parent WES studies in research and diagnostics for neurodevelopmental disorders[47], accelerating disease gene identification and increasing the diagnostic yield for these disorders. The data presented here suggest that a similar benefit could be achieved from trio-based exome sequencing in male infertility. In order to achieve this there is an urgent need to expand on this work as larger studies are essential to identify recurrently DNM genes and further demonstrate the exact contribution of DNMs to male infertility. Modeling studies recently done for developmental disorders showed that more than 350k trios may be required to have approximately 80% power to detect all haploinsufficient genes causing this disorder[48]. Evidently, these numbers can only be reached by implementing trio-based exome sequencing as a routine diagnostic test and by sharing these diagnostic data with the international research community. This research community

**Table 2 Rare pathogenic missense mutations in exome data from various cohorts of infertile men and fertile control cohorts.**

| Gene | Missense Z-score | Total infertile men (n = 2,506) | Fertile Dutch men (n = 5,784) | Burden test infertile vs. fertile men | |
|---|---|---|---|---|---|
| | | | | p value | Adjusted p value |
| RBM5 | 4.17 | 7 | 0 | 0.0002 | 0.03 |
| HUWE1 | 8.87 | 6 | 0 | 0.001 | 0.12 |
| REN | 0.80 | 7 | 1 | 0.001 | 0.21 |
| HIST1H1D | −8.06 | 10 | 5 | 0.004 | 0.59 |
| ABLIM1 | 1.62 | 6 | 1 | 0.004 | 0.60 |
| FUS | 2.21 | 4 | 0 | 0.01 | 1 |
| CNOT4 | 3.49 | 5 | 1 | 0.01 | 1 |
| CDC5L | 2.78 | 6 | 2 | 0.01 | 1 |
| ZNF629 | 3.86 | 6 | 2 | 0.01 | 1 |
| PCDHB1 | 1.02 | 11 | 8 | 0.01 | 1 |
| AK3 | −1.97 | 10 | 7 | 0.02 | 1 |

Genes affected by a rare missense DNM were investigated in additional cohorts of infertile patients and a cohort of verified fertile men to identify other individuals carrying rare missense mutations. A burden test was used to compare the total number of predicted pathogenic missense mutations observed in the infertile vs. fertile men. A two-tailed Fisher's Exact test was performed with and without Bonferroni correction applied to adjust p values for multiple testing of all 152 genes of interest. (Also see Supplementary Data 5).

will also have the enormous task of functionally validating the impact of these DNMs on spermatogenesis. Altogether, this will not only help to increase the diagnostic yield for men with infertility but will also enhance our fundamental biological understanding of human reproduction and natural selection. In addition, it will indicate whether male infertility follows a dominant inheritance pattern, and this has impact for disease transmission. Couples that seek treatment for male infertility should be counseled on the risk of transmitting this condition to their offspring, something that is now limited to couples receiving fertility treatment due to Y-chromosome deletions. Male infertility is also increasingly seen as the most visible symptom of a more complex disease with associated comorbidities[49]. Studying the long-term health of men with DNMs in specific genes should help in identifying genotype–phenotype correlations that may impact more than the fertility of these men.

## Methods

**Cohort of infertile patients and fertile parent trios**. We enrolled a total of 185 patients who presented with unexplained (idiopathic) azoospermia (N = 111) or severe to extreme oligozoospermia (with or without asthenozoospermia N = 74) at the Radboudumc outpatient clinic between July 2007 and October 2017 (N = 170) and at the Newcastle upon Tyne Hospitals NHS Foundation Trust (Newcastle, UK) between January 2018 to January 2020 (n = 15). The reference values and semen nomenclature were used according to the WHO guidelines[50] (see Supplementary Note). Clinical evaluation did not lead to an etiologic diagnosis and all patients were negative for AZF deletions and chromosomal anomalies (see Supplementary Notes). The study protocol was approved by the respective Ethics Committees/ Institutional Review Boards (Nijmegen: NL50495.091.14 version 5.0, Newcastle: REC ref. 18/NE/0089) and written informed consent from all patients and their parents was obtained prior to enrollment in the study. We used residual genomic DNA extracted from a blood sample taken at the time of evaluation and treatment at the fertility center. DNA from all proband's parents was obtained from saliva by using the Oragene OG-500 kit (DNA Genotek, Ottawa, Canada).

**Immunofluorescence staining of human testis biopsies**. Tissue sections were cut from formalin fixed paraffin embedded (FFPE) testicular biopsies. As staining controls testicular biopsies obtained from fertile men after a previous vasectomy was used. FFPE sections were prepared for staining following standard protocols. To detect RBM5, antibody HPA018011 from Atlas Antibodies was used in a 1:500 dilution. For detection, a donkey anti-rabbit Alexa488 antibody was used (A-21206, 1:1000 dilution), which was applied in combination with lectin coupled to Alexa568 (L32458, 1:1500 dilution) to detect the acrosome (both Thermo Scientific). Slides were counterstained with DAPI. Images were obtained with a Zeiss Axio Imager ZI fluorescence microscope equipped with the Zen software package.

**Cohort of verified fertile Dutch parents**. We used an anonymized exome dataset derived from 5784 Dutch men and 5803 Dutch women who had conceived at least one child as a control cohort for the frequency of rare variants in fertile men and fertile women. These men and women received routine exome sequencing at the

Radboud diagnostics center as the healthy parent of a child with a severe illness. Although these men fathered a child with intellectual disability, their fertility is expected to be similar to an unselected sample of the male population.

**Exome sequencing**. WES samples were prepared and enriched following the manufacturer's protocols of either Illumina's Nextera DNA Exome Capture kit or Twist Bioscience's Twist Human Core Exome Kit. All sequencing was performed on the NovaSeq 6000 Sequencing System (Illumina) achieving comparable results covering more than 99% of all exonic regions using either kit (Supplementary Table 2) and an average depth of 72× (Illumina's Nextera Kit) and 99x (Twist Bioscience's Kit) (see Supplementary Fig. 9 and Supplementary Table 3). Sequenced reads were aligned to Human Reference Genome (GRCh37.p5/hg19) using BWA-Mem v0.7.17[51], Picard[52], and GATK v4.1.4.1[53]. The sex, ancestry and relatedness of each samples was calculated using peddy[54], samples found to have the incorrect sex or were unrelated to the correct samples were excluded from this study. Following best practice recommendations, single nucleotide variations and small indels were identified and quality-filtered using GATK's HaplotypeCaller obtaining comparable results independently of the kit or the origin of the DNA (see Supplementary Table 3). Afterwards, all variants were further analysed using a custom GATK4-based algorithm to identify and separate high- and low-confidence de novo variants from inherited variants. Briefly, posterior genotype probabilities (GQ) were recalculated for each sample at each variant site using Bayes' rule to take into account family and population priors[53,55]. Proband variants absent in parental samples with recalculated proband GQ > = 10 and allele count (AC) below 4 or allele frequency (AF) < 0.1% in all samples, whichever is more stringent, were classified as low-confidence DNMs. Variants with recalculated GQs ≥ 20 and the same AC/AF criterion were classified as high confidence DNMs. Afterwards, tagged variants with coverage <10, variant read percentage <15% and GATK quality scores <400 were removed to ensure only the most reliable variants were considered. Sanger sequencing was then used to validate DNMs calls. Ensembl's Variant Effect Predictor (VEP)[56] was used to fully annotate all de novo variants.

**Variant filtration and interpretation**. The primary stages in filtering of variants included removing all variants with an allele frequency of >0.1% in the gnomAD database to only include rare variants in our analysis. All variants then with <10 reads in the exome data and/or less than 15% of these reads containing the mutation were then removed. At this stage, any remaining variants lying outside the exonic regions were then removed. This provided the initial list of 192 rare de novo variants. All synonymous and non-protein-altering spice site variants were then removed, leaving a total of 145-protein-altering rare DNMs. Pathogenicity prediction was then based on SIFT[20], MutationTaster[21], and PolyPhen2[22] and all variants were classified according to the American College of Medical Genetics and Genomics (ACMG) and the Association for Molecular Pathology (AMP) 2015 guidelines[57]. All protein-altering variants predicted to be pathogenic by at least 2 out of 3 prediction models, absent from the fertile male cohort, present in <5 males in the gnomAD database were considered for further functional analysis (n = 84). Maternally inherited mutations present in genes identified as having a protein-altering DNM were identified in all patients and submitted to the exact same method of filtration and interpretation as described above.

Functional analysis was split into six different categories, each category provided a score of either 1 or 0 depending on whether they met the threshold for that category. These categories included: RNA expression of the gene in the testis, RNA enrichment in the testis or presence in spermatogenesis, protein expression in the testis, whether an infertile mouse model already exists for the given gene, the

protein function in relation to spermatogenesis and finally whether the given gene interacts with any known fertility genes. For expression levels retrieved for each gene of interest from the GTEx database (https://www.gtexportal.org/), an expression of medium (≥10 < 100 TPM) or high (>100 TPM) gave a score of 1 with low (>2 < 10 TPM) and no expression (<2 TPM) giving a score of 0. RNA enrichment was based on elevated expression (tissue enriched, group enriched, or tissue enhanced) in the Human Protein Atlas[58] or being among the genes up- or downregulated during spermatogenesis as found in a recent single cell RNA sequencing study[59]. Protein expression was retrieved from the Human Protein Atlas[58] and interaction with known infertility genes[3] was calculated using STRING version 11[24]. The final classification of the genes was then split into Not causative, Unlikely causative, Unclear and Possibly causative. These classifications were given based on the variant scores out of 6 with: [0 points + not expressed/not detected/not present on several occasions = Unlikely causative], [0 points + "Unknown" on several occasions = Unclear], [1–2 points = Unclear] and [3–6 points = possibly causative].

**CNV analysis**. CNV calling was performed on our trio-based exome data with a custom GATK4-based pipeline. This workflow exploits the GATK4 sequence read depth normalization[60] and a custom R based segmentation and visualization[61]. Parental samples from the trios under examination were used as controls for the normalization step. The CNVs detected were annotated using AnnotSV (https://lbgi.fr/AnnotSV/)[62]. CNVs present in more than 1% of the samples of the Database of Genomic Variants present in more than 10% of the patients were excluded from the analysis. The remaining rare deletions and duplications were individually inspected through the genomic profiles and detailed Log2Ratio plots generated by the workflow. Only CNVs involving more than 2 exons were further considered to minimize the inclusion of false positives, and we selected 2 CNVs present in the probands but absent in their parents for further validation.

**Variant validation**. Validation of low-quality DNMs was performed using standard Sanger sequencing approach on an Applied Biosystems SeqStudio Genetic Analyzer (ThermoFisher, MA, USA) to confirm the presence of the mutation in probands and its absence in the parents. Primers for each SNV were designed using PrimerZ[63] (Supplementary Data 8) and PCR reactions were performed using AmpliTaq 360 DNA Polymerase (ThermoFisher, MA, USA) according to the manufacturer's protocol.

Validation of CNVs was performed with the whole genome Illumina Infinium CytoSNP-850K v1.1 microarray platform for the larger deletion on chromosome 11 and a gene-specific TaqMan Copy-Number assay designed for *NXT2* was exploited to validate the smaller CNV using the Applied Biosystems QuantStudio 7 Flex Real-Time PCR System (ThermoFisher, MA, USA).

**Functional enrichment**. To evaluate the intolerance of each gene for loss-of-function (LoF) mutations, we used the probability of LoF intolerance (pLI) score, based on data from the Genome Aggregation Database (gnomAD)[8] containing genetic data from 141,456 individuals. We computed the likelihood of the observed median pLI score of each gene (LoF in controls) set compared to the expected median pLI based on the method described in Lelieveld et al.[64]. In short, we simulated the expected number of recurrently mutated genes by redistributing the observed number of mutations at random over a determined set of genes based on their specific LoF and functional mutation rates, however, in contrast to Lelieveld et al.[64] and Samocha et al.[23] before them instead of using the complete set of 18,226 pLI annotated genes to obtain expected median pLI scores, we used a set of 2766 coding DNMs in 1941 control individuals, downloaded from the de novo-db version 1.6.1 (http://de novo-db.gs.washington.edu/de novo-db/)[12], to correct for the gene-specific mutation rate. The empirical *P* value was calculated by comparing the observed median pLI to the expected pLI following 100,000 random sampling simulations. Case and fertile controls were processed using the exact same filtration and annotation parameters as described above so that each variant detected was evaluated in a comparable manner. The same method was then repeated using the Loss-of-function observed/expected upper bound fraction or LOEUF score, which also is an indicator of LoF intolerance. To evaluate the impact of the de novo missense mutations to each gene, we used missense Z-scores calculated by gnomAD[10,23] to predict the tolerance of each gene to variation in place of the pLI scores when applying the Lelieveld et al.[64] methodology described above following 100,000 simulations. The presence of missense mutations in intolerant genes was compared between predicted pathogenic and benign using a two-tailed Mann–Whitney U test in our samples and in controls independently. To predict the affected protein function and the potential role in disease, we evaluated the interactions between the genes with a DNM using STRING version 11[24].

**Additional cohorts of infertile men**. The strongest candidate genes with DNMs were further investigated in exome data from four additional cohorts of infertile men. For the Italian cohort of 48 patients with NOA, exome sequencing was carried out as a service by Macrogen Inc. (Republic of Korea) utilizing the Agilent SureSelect_V6 enrichment and a NovaSeq 6000. The German Male Reproductive Genomics (MERGE) study comprised exome data of 887 men with azoo-, crypto-, or severe oligozoospermia. Known causes for male infertility like chromosomal

aberrations and microdeletions of the AZF region were excluded in advance. WES was performed as previously described[65]. The 88 patients diagnosed with male infertility participating in the Geisinger-Regeneron DiscovEHR collaboration were selected from deidentified EHR information using the ICD-10CM code N46 which refers to "Male Infertility" including oligospermia, azoospermia, other male infertility and male infertility unspecified. All patients were sequenced at the Regeneron Genetics Center (RGC) as previously described[39]. In brief, 1ug of genomic DNA per sample was used for targeted exome capture using the NimbleGen VCRome 2.1 or the IDT XGen reagents. Captured libraries were sequenced on the Illumina HiSeq 2500 platform with v4 chemistry using paired-end 75 bp reads. Exome sequencing was performed such that >85% of the bases were covered at 20× or greater. Raw sequence reads were mapped and aligned to the GRCh38/hg38 human genome reference assembly using BWA-mem[51], single nucleotide and indel variants were called using GATK's HaplotypeCaller[53]. The Genetics of Male Infertility INitiative (GEMINI) is a multicenter study funded by the United States NIH. The GEMINI project performed whole-exome sequencing on 1,011 unrelated men diagnosed with spermatogenic failure, the vast majority with unexplained NOA. Sequencing of genomic DNA was performed at the McDonnell Genome Institute of Washington University in St. Louis, MO, USA, using an in-house exome targeting reagent capturing 39.1 Mb of exome and 2 × 150 bp paired-end sequencing on Illumina HiSeq 4000. Following sample QC, a final cohort of 924 men were analysed as part of the current study.

Genetic variants identified within the 152 candidate genes were extracted from each exome dataset. Consistent with our filtering method described above variants with <10 reads and/or <15% reads containing the mutation were discarded. To minimize discrepancies between genomic positions and annotation, genomic coordinates were recalculated to the GCRh37/hg19 where necessary and fully reannotated with VEP[56]. Following annotation variants from each of the additional case and control cohorts were filtered and processed in an identical manner as previously described. Shortly, variants with allele frequency >1% in gnomAD were discarded to focus only on rare variants: Pathogenicity predictions based on SIFT[20], MutationTaster[21] and PolyPhen[22] were then used to exclude benign variants, all remaining variants were classified according to ACMG guidelines[56]. Like before, all protein-altering variants were considered pathogenic if predicted to be so by at least 2 out of 3 prediction models, absent from the fertile cohorts and present in <5 males in the gnomAD database. A similar analysis was done for all variants obtained in the control cohorts. However, just to be clear, for the male control cohort we did not exclude variants as pathogenic if they were present in this cohort itself.

**Burden testing**. Having identified several likely pathogenic rare loss-of-function and missense mutations in these 152 genes we performed a gene-based burden test to compare the combined data in all cohorts of infertile men with the control cohort of fertile fathers. The proportion of individuals with pathogenic variants in each of the 152 genes was statistically evaluated using two-tailed Fisher's Exact tests, individual *p* values were corrected using the Bonferroni method corrections to adjust for performing 152 consecutive statistical tests and reduced the risk of Type I errors. Similarly, a gene-based burden test was performed to compare fertile fathers with fertile mothers from the control cohort of verified fertile parents to investigate whether any of the sexes predominantly carried a greater number of rare pathogenic mutations.

**Phasing analysis to determine parent-of-origin**. The origin of DNMs identified in the exomes of patients was first investigated in the short-read data by performing phasing analysis on those variants that contained a parental informative SNP (iSNP) within 150 bp from the DNM. As a next step, all DNMs were target-enriched with long-range PCR and sequenced using the Oxford Nanopore's MinION sequencer (Oxford Nanopore technologies, Oxford Science Park, UK). Target regions were designed to encapsulate both the DNM and a parentally informative SNP, from which parent-of-origin, and allele frequencies (percentage read counts associated to a given allele) could be ascertained and DNM pre-/post-zygosity could be determined.

Primers were designed using Primer3[66] (version 2.3.6) and GRCh37.p5 based in-house GUI-wrapped pipeline. All expected fragment sizes were limited to a maximum of 12 kb for quality control and enrichment success rate. For those DNMs with no exome supported iSNPs within a 10 kb distance, primers were designed to cover approximately 2.5 kb on either side of the DNM with the expectation of finding additional iSNPs in the intronic regions. Long-range PCR target enrichment was carried out using our optimized running conditions of 3 separate supermixes/enzymes (see Supplementary Table 4). Sample fragment sizes were confirmed using gel electrophoresis, and quantities were measured with the Qubit dsDNA HS kit (ThermoFisher Scientific, Waltham, MA, USA), with the best quality supermix enrichment for each given sample/target selected for sequencing, where quality was assessed by cleanest banding in gel electrophoresis and greatest concentration.

The long-range PCR target enrichments of >20 ng were prepared for sequencing with the ONT ligation sequencing kit (SQK-LSK109) following the manufacturer's protocol, with adjustments for sample type and yield. Individual sample libraries were concentrated where necessary at given bead clean-up steps and pooled based on fragment size. Fragment size-based pools were combined prior to flowcell loading. Prepared samples were sequenced on the MinION using the FLO-MIN106

version 9.4 flowcell platform. Flowcells were run until complete pore exhaustion, with minimal refuel of flowcells performed whenever active pore percentages dropped below 70%, achieving an average of 30 billion basecall yields per flowcell and coverage depth per sample of >5000×.

The sequence signal data in multi-fast5 format were basecalled using Guppy[67] (version 3.4.4, https://nanoporetech.com/), resulting fastq outputs were adapter trimmed and low-quality reads discarded using cutadapt (version 2.5)[68]. Cleaned fastq files were mapped against Human Reference Genome (GRCh37.p5/hg19) using BWA-Mem[51] (version 0.7.17), and sample targets were extracted from the resulting BAM file using SAMtools (version 0.1.19)[69]. Aligned ONT reads were phased using an in-house tool, with frequencies and pre/post-zygosity calls affirmed via IGV and principal component analysis using the available exome sequence data for probands and parents to support the ONT data.

**Reporting summary**. Further information on research design is available in the Nature Research Reporting Summary linked to this article.

## Data availability

Sequencing data has been deposited in the European Genome-phenome Archive(EGA) under the accession code EGAS00001005417 and will be made available upon reasonable request for academic use and within the limitations of the provided informed consent by the corresponding author upon acceptance. Every request will be reviewed by the Newcastle University Male Infertility Genomics Data Access Committee; the researcher will need to sign a data access agreement after approval.

## Code availability

The code for the integrated pipeline used to process sequencing data to detect and call rare germline copy-number variants (CNVs) is available at https://github.com/AnetaMikulasova/CNVRobot.

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

## Acknowledgements
We are grateful for the participation of all patients and their parents in this study. We thank Laurens van de Wiel (Radboudumc), Sebastian Judd-Mole (Monash University), Arron Scott and Bryan Hepworth (Newcastle University) for technical support, and Margot J Wyrwoll (University of Münster) for help with handling MERGE samples and data. This project was funded by The Netherlands Organization for Scientific Research (918-15-667) to J.A.V. as well as an Investigator Award in Science from the Wellcome Trust (209451) to J.A.V. a grant from the Catherine van Tussenbroek Foundation to M.S.O. a grant from MERCK to R.S. a UUKi Rutherford Fund Fellowship awarded to B.J.H. and the German Research Foundation Clinical Research Unit "Male Germ Cells" (DFG, CRU326) to C.F. and F.T. This project was also supported in part by funding from the Australian National Health and Medical Research Council (APP1120356) to M.K.O.B., by grants from the National Institutes of Health of the United States of America (R01HD078641 to D.F.C. and K.I.A., P50HD096723 to D.F.C.) and from the Biotechnology and Biological Sciences Research Council (BB/S008039/1) to D.J.E.

## Author contributions
This study was designed by M.S.O., L.E.L.M.V., L.R. and J.A.V. R.M.S., J.G., H.T. and G.W.v.d.H. provided all clinical data and performed the TESE histology and cytology analysis under supervision of L.R., D.D.M.B., E.S., K.F., K.D.H. and K.M. J.C. performed the exome sequencing with support from B.A. and bioinformatics support was provided by M.J.X., G.A., C.G. and S.C. Sanger sequencing was performed by P.F.d.V., H.I., H.E.S., L.E.B. and B.K.S.A. M.S.O. and H.E.S. performed the SNV analyses with support from M.J.X., F.K.M. performed CNV analysis with support from A.M. and M.S.K. and G.S.H. and L.E.B. performed the phasing. D.J.E., H.S., B.J.H. and M.K.O.B. provided support on the functional interpretation of mutations. D.F.C., L.N., C.F., S.K., F.T., K.I.A., A.R.E., C.K. and C.G.-J. were involved in the replication study. The first draft of the paper was prepared by M.S.O., H.E.S., R.M.S., M.J.X., G.W.v.d.H. and J.A.V.. All authors contributed to the final paper.

## Competing interests
The authors declare no competing interests.

## Additional information

[1]Department of Human Genetics, Donders Institute for Brain, Cognition and Behaviour, Radboudumc, Nijmegen, The Netherlands. [2]Department of Obstetrics and Gynaecology, Radboudumc, Nijmegen, The Netherlands. [3]Biosciences Institute, Faculty of Medical Sciences, Newcastle University, Newcastle upon Tyne, UK. [4]School of BioSciences, Faculty of Science, The University of Melbourne, Parkville, VIC, Australia. [5]Department of Genetic Medicine, The Newcastle upon Tyne Hospitals NHS Foundation Trust, Newcastle upon Tyne, UK. [6]Foundation for Research in Genetics and Endocrinology, Institute of Human Genetics, Ahmedabad, India. [7]Department of Human Genetics, Radboud Institute for Molecular Life Sciences, Radboudumc, Nijmegen, The Netherlands. [8]Division of Human Genetics, Center for Biomedical Research, Faculty of Medicine, Diponegoro University, Semarang, Indonesia. [9]Newcastle Fertility Centre, The Newcastle upon Tyne Hospitals NHS Foundation Trust, Newcastle upon Tyne, UK. [10]Department of Cellular Pathology, The Newcastle upon Tyne Hospitals NHS Foundation Trust, Newcastle upon Tyne, UK. [11]Genomics Core Facility, Faculty of Medical Sciences, Newcastle University, Newcastle upon Tyne, UK. [12]Bioinformatics Support Unit, Faculty of Medical Sciences New, castle University, Newcastle upon Tyne, UK. [13]Department of Urology, Radboudumc, Nijmegen, The Netherlands. [14]Department of Pathology, Radboudumc, Nijmegen, The Netherlands. [15]Division of Genetics, Oregon National Primate Research Center, Oregon Health & Science University, Beaverton, OR, USA. [16]Institute of Reproductive Genetics, University of Münster, Münster, Germany. [17]Centre of Reproductive Medicine and Andrology, Department of Clinical and Surgical Andrology, University Hospital Münster, Münster, Germany. [18]Department of Surgery, Division of Urology, University of Utah School of Medicine, Salt Lake City, UT, USA. [19]Andrology Department, Fundació Puigvert, Universitat Autònoma de Barcelona, Instituto de Investigaciones Biomédicas Sant Pau (IIB-Sant Pau), Barcelona, Catalonia, Spain. [20]Department of Biomedical, Experimental and Clinical Sciences "Mario Serio", University of Florence, Florence, Italy. [21]Regeneron Genetics

Center, Tarrytown, NY, USA. [42]These authors contributed equally: M. S. Oud, R. M. Smits, H. E. Smith. [43]These authors jointly supervised this work: M. J. Xavier, G. W. van der Heijden, J. A. Veltman. *A list of authors and their affiliations appears at the end of the paper.
✉email: joris.veltman@newcastle.ac.uk

## Genetics of Male Infertility Initiative (GEMINI) consortium

Donald F. Conrad[22], Liina Nagirnaja[22], Kenneth I. Aston[23], Douglas T. Carrell[23], James M. Hotaling[23], Timothy G. Jenkins[23], Rob McLachlan[24,25], Moira K. O'Bryan[4], Peter N. Schlegel[26], Michael L. Eisenberg[27], Jay I. Sandlow[28], Emily S. Jungheim[29], Kenan R. Omurtag[29], Alexandra M. Lopes[30,31], Susana Seixas[30,31], Filipa Carvalho[30,32], Susana Fernandes[30,32], Alberto Barros[30,32], João Gonçalves[33,34], Iris Caetano[33], Graça Pinto[35], Sónia Correia[35], Maris Laan[36], Margus Punab[37], Ewa Rajpert-De Meyts[38], Niels Jørgensen[38], Kristian Almstrup[38], Csilla G. Krausz[39,40] & Keith A. Jarvi[41]

[22]Department of Genetics, Oregon National Primate Research Center, Oregon Health & Science University, Beaverton, OR, USA. [23]Andrology and IVF Laboratory, Department of Surgery (Urology), University of Utah School of Medicine, Salt Lake City, UT, USA. [24]Hudson Institute of Medical Research and the Department of Obstetrics and Gynaecology, Monash University, Clayton, VIC, Australia. [25]Monash IVF and the Hudson Institute of Medical Research, Clayton, VIC, Australia. [26]Department of Urology, Weill Cornell Medicine, New York, NY, USA. [27]Department of Urology, Stanford University School of Medicine, Stanford, CA, USA. [28]Department of Urology, Medical College of Wisconsin, Milwaukee, WI, USA. [29]Washington University in St Louis, School of Medicine, St Louis, MO, USA. [30]i3S—Instituto de Investigação e Inovação em Saúde, Universidade do University of Porto, Porto, Portugal. [31]IPATIMUP—Instituto de Patologia e Imunologia Molecular da Universidade do Porto, Porto, Portugal. [32]Serviço de Genética, Departamento de Patologia, Faculdade de Medicina da Universidade do Porto, Porto, Portugal. [33]Departamento de Genética Humana, Instituto Nacional de Saúde Dr Ricardo Jorge, Lisboa, Portugal. [34]ToxOmics, Faculdade de Ciências Médicas, Universidade Nova de Lisboa, Lisboa, Portugal. [35]Centro de Medicina Reprodutiva, Maternidade Dr. Alfredo da Costa, Lisboa, Portugal. [36]Institute of Biomedicine and Translational Medicine, University of Tartu, Tartu, Estonia. [37]Andrology Center, Tartu University Hospital, Tartu, Estonia. [38]Department of Growth and Reproduction, Rigshospitalet, University of Copenhagen, Copenhagen, Denmark. [39]Department of Experimental and Clinical Biomedical Sciences, University of Florence, Florence, Italy. [40]Andrology Department, Fundacio Puigvert, Instituto de Investigaciones Biomédicas Sant Pau (IIB-Sant Pau), Barcelona, Spain. [41]Division of Urology, Department of Surgery, Mount Sinai Hospital, University of Toronto, Toronto, ON, Canada.

