## [Peer Review File · Nature Communications]

A de novo paradigm for male infertilityREVIEWER COMMENTS

Reviewer #1 (Remarks to the Author):

Comments for the Author:

Oud MS et al. performed exome sequencing in 185 trios of infertile male and their unaffected parents and identified 145 rare protein-altering DNMs and two de novo deletion CNVs. Although the sample size is relatively small compare to other well-studied diseases, however, it's still a valuable resource giving the rarity of this type of data to this disease and the challenges of collecting such a cohort. They classified 29 of the 145 DNMs as possibly causative to the phenotype. For one known gene, RBM5, they identified 5 additional rare pathogenic missense mutations in additional infertile patients with a significant burden compared to the fertile cohort. Overall, this study presented an interesting dataset with a relatively larger sample size than what's recently published (n=13 trios), but further analyses are needed to build a solid association of the DNMs and genes to severe male infertility. My other detailed comments are as follows:

Major comments:

1. Abstract:

1) Novelty of the study: A trio-based exome study was published has also been cited, the author cannot neglect it just because of the small sample size. The following statement in the abstract needs to be reworded "Our results provide the first evidence for the role of DNMs in severe male infertility" in the abstract. And in lines 151-153 on page 6, "For replication purposes, unfortunately no other trio-based exome data are available for male infertility, although we note that a pilot study including 13 trios was recently published", this sentence is conflicted itself and also needs to be reworded. And compare the DNMs even only has 13 trios there.

2) New candidate genes: The authors need to be specific on the exact number of new candidate genes been identified, by saying "point to many new candidate genes affecting fertility" in the abstract is not a precise way to present important results and the takeaway for readers.

2. Gene intolerance and pathogenicity: The author applied both pLI and mis_Z scores to assess the causative of genes with DNMs identified. There are multiple issues here:

1) Gene intolerance score is only a prediction, which has pitfall of itself (i.e. some known disease genes have low pLI score or even almost 0, see PMID: 30977936). It's helpful as can provide more support, but

not good to use it as direct evidence of pathogenicity. In addition, the authors should also consider including the RVIS score and the LOEUF score.

2) The authors performed different comparisons for pLI and mis_Z scores but simply say “We observed a significant enrichment of Loss-of-function(LoF) DNMs in LoF-intolerant genes (p-value= 1.00×10^{-5}) as well as predicted pathogenic missense DNMs in missense-intolerant genes (p value= 5.01×10^{-4})” in one sentence in the abstract. This is misleading and the analyses are problematic. For pLI, the authors compared the median pLI score of genes with LoF DNM in their cohort (n=17) with a public control cohort (n=181), the number of DNMs differ so much in two sets, a direct comparison of the median pLI here could be biased. For mis_Z, the authors compared the predicted pathogenic missense DNMs (n=63) with benign ones (n=59) but both from their trios. The significance here is not surprising and doesn't tell much. It's kind of a “cherry-picking” comparison as you are comparing the pathogenic missense to benign ones you have already known. Instead, the comparison between overall missense DNMs in patients and control populations shows no significant difference as in Supplementary Figure 4.

3) First of all, the authors need to apply the same filtering and re-annotate the control DNMs the same way as for their cohort to ensure a consistent and unbiased comparison. A different variant could be annotated with different effects by a different method or if on different isoforms. And providing a supplementary table of the variants and the scores used in the analysis is needed.

4) Figure 1 is unnecessarily complicated. The figure presented comparisons that were not mentioned in the paper. Suggest simplifying it to show only the important message. And Figure 2 is too simple to be as an independent figure, suggest combining it with Figure 1 as both are about gene intolerance.

3. QC of exome sequencing: As different exome capture kit and DNA source across the samples, the authors need to provide more details in supplementary on sequencing QC, at least but not limited to the following:

1) A plot of the average target coverage on all samples, indicate in different colors or shapes indicating the capture kit and DNA source (saliva vs blood).

2) Compare the capture regions of the two kits, if there any critical genes or regions in one but not another.

3) Comparison of variant calling on saliva and blood DNA.

4. DNMs identification and filtering:

1) The filtering of the candidate mutation on allele balance ($AB > 0.15$) is a bit low, which could introduce somatic mutations and need a careful assessment. The methods (line 39 in “Exome sequencing”) mentioned “Variants absent in parental samples”, is this mean the parents have no alternative allele ($AB=0$), or just with $AB < 0.15$ that not been classified as variants per the authors’ filtering criteria? This needs to be further specified.

2) Supplementary Table 1 needs to display in an Excel spreadsheet for easier reuse of the data. The authors also need to provide additional columns for the reference allele and alternative allele, providing the parent-child read depth (DP), allele count (AC), and allele balance (AB) will be also important to increase the future usage of the data.

5. Analysis in additional cohorts: When searching for rare LoF and predicted pathogenic missense mutations in additional infertile and fertile cohorts, the authors should check for all genes with DNM in the study cohort, or at least for the 145 protein-coding DNM genes, but not only the 33 top genes.

6. Beyond DNMs: The authors should also look for candidates harboring rare maternally inherited variants, autosomal recessive variants, and X-chromosome hemizygous variants in addition to the DNMs to increase diagnostic yield. Similarly, for CNVs, it is a bit low as there are only 2 de novo CNVs identified. Are there any rare maternally inherited CNVs that harboring any male infertile genes that could explain the patient’s phenotype?

7. Statistical analyses: The authors need to provide more details on the statistical comparisons applied in the study, like what kind of statistical method was used; and if multiple test correction was applied, the details of how many tests have been corrected should be specified other than just simply say Bonferroni correction was applied, as this could affect the adjusted p-value so as the significance. The authors should provide both the raw and adjusted p-value wherever applied.

Minor comments:

1. The paper needs to be re-formatted based on the journal style, the language needs to double-check carefully. For example, the gene names should be consistently in italic; gnomAD was wrongly written as “GnomAD”; in methods section “Variant filtration and interpretation” line 64 “Tissue expression was retrieved from the Project The final classification of the genes ...” doesn’t make sense, maybe missed something.

2. Table 2 and supplementary table 2 are very hard to follow, readers could get lost with the many columns. Suggest to reorganize or simplify for easy follow, e.g. the additional cohorts are not necessary to display in individual columns, a sum column will work fine here.

3. Variant phasing: The authors need to be clear and specific about the number of DNMs been phased in the 2nd paragraph on page 7.

4. Some of the references need to be updated to the correct publication (e.g. denovo-db).

5. The actual rare LoF and missense variants and details worth presenting in a supplementary table, only show counts is not helpful for future data reuse; so as for the DNMs from the control cohort been used in the comparison, and also a good place to provide the underlying intolerance scores.

Reviewer #2 (Remarks to the Author):

The authors hypothesize that de novo mutations play an important role and explain a significant fraction of genetic causes of male infertility. Exome sequencing was performed in 185 trios and identified 192 DNMs, 145 of which were rare protein altering DNMs, followed by systematic analysis of these variants. The authors reported 29 rare protein altering DNMs were possibly causative of male infertility. The study aims to provide evidence of the role of DNM in severe male infertility and discovery of new candidate genes. It would add to the contribution of this manuscript were the authors explain the potential clinical utility on how these findings will impact current care of male infertility.

Comments:

The authors discovered 192 rare DNMs, of which 145 alter protein coding and 29 affected genes were linked to male reproduction and classified as possibly causative of the male infertility phenotype. Evidence from systematic analyses implies a burden of predicted pathogenic and missense DNMs in LoF-intolerant genes in infertile men. It is difficult to obtain direct genotype-phenotype correlations and establish autosomal dominant pathogenic mechanisms, especially when many genes have only a single variant. Might the authors elaborate on what kind of future work must be carried out in order to solidify the evidence to translate the findings to benefit clinical diagnoses?

It is not clear how the authors arrive at the final classification of the gene variants in the materials and methods. How much weight is put into evidence each from in-silico pathogenicity prediction tools, ACMG guidelines, functional analyses?

It should be noted that the small 6kb de novo CNV in Proband_039 is probably not associated with male infertility.

The authors should explain possible reasons for candidate “rare predicted pathogenic missense variants” or “possibly causative variants” in genes found also in fertile Dutch Men. The authors should discuss the limitations and potential false discovery of causative variants using their approach. For example, SOGA1 is listed in Table 2 as rare potential pathogenic missense mutation but there are also 15 variants in fertile Dutch men. Possibly causative DMN variants in CDK5RAP2 (Table 1) are also found in fertile Dutch men.

Was confirmation of biological parents performed for all trios?

The results of this study are not the first evidence of the role of DNMs in male infertility. Also mentioned by the authors, a pilot study of 13 cases using trio-based exome analysis reported DNMs and candidate genes in idiopathic male infertility was recently published.

As a study of DNMs as a contribution to idiopathic male infertility, it is surprising that 52 men (28%) had a family history of genetic abnormalities or infertility. Did the authors also investigate the landscape of inherited pathogenic variants? What could be possible reasons for the high incidence of family history of genetic abnormalities or infertility?

Minor comments:

p. 3 line 71 Correct “explaining” to “explain”.

p. 5 line 114 Correct “highlights” to “highlight”.

p. 5 line 128 Insert a comma following RBM5.

Supplementary Figure 7 It is atypical to indicate the father on the right and the mother on the left. Please redraw.

1.1 Supplementary notes line 29 Delete “with” at end of the line.

1.1 Supplementary notes line 63 Restate “...and the descent of four men was...”

1.1 Supplementary notes line 66 Insert “for” following “patients” and replace “was” with “were.”

1.1 Supplementary notes line 74 Delete the second use of “be”.

1.1 Supplementary notes line 89 Correct “was” to “were”.

1.1 Supplementary notes line 98 Replace “let” with “left”.

1.1 Supplementary notes line 126 Replace “show’ with “shows”.

1.1 Supplementary notes line 134 Add a period at the end of the sentence.

1.1 Supplementary notes line 136 Change “cells” to “cell” and “have” to “is”.

1.1 Supplementary notes line 148 Change “that’ to “who”.

Supplementary Figure 2 Correct “atrofy” to “atrophy”, and delete “?” after 50% and after 20%.

Materials and methods:

line 17 Change “was” to “were”.

line 40 Insert a comma after “stringent”.

line 64 Correct partial sentence, “Tissue expression was retrieved from the Project...”.

line 70 Use lower case for sequencing.

line 84 Use lower case for expected.

line 88 Use upper case for Supplementary.

line 100 Is “basecalled” a verb?

line 113 Replace “Variations” with “Variants”.

lines 122 and 129 Use lower case for “g” in gnomAD.

line 130 Replace “Chi-Squared test” with “Chi-square test”.

line 152 Delete comma following “Nagrinja” and insert a comma following “al.”.

line 164 Change “Consistently” to “Consistent” and insert a comma following “above”.

line 166 Change “coordinated” to “coordinates”.

line 181 Change “is” to “are” and “requires” to “require”.

Table of Mouse models:

Use italicized gene symbols throughout.

Provide criteria for classification of “Conclusion variant/gene” in table legend.

Supplementary Table 2 Use upper case for Table and italicize gene symbols.

Supplementary Table 3 line 11 Change “on” to “in” prior to both instances of RBM5, delete “the” prior to Proband”, and italicize second instance of RBM5.

Supplementary Table 3 line 12 Italicize RBM5.

Supplementary Table 3 line 13 Change “no” to “not”.

Supplementary Table 5 Use upper case for Table.

Table 1 – Clinical data

Proband_028 Correct “fathers” to “father’s”.

Proband_039 Correct “familie” to “family” and “mothers” to “mother’s”.

Proband_066 Correct “mothers” to ‘mother’s” in two instances.

Proband_129 Correct “mothers” to ‘mother’s”.

Proband_145 Correct “Sisters” to “Sister”.

Proband_161 Correct “irregulair” to “irregular”.

Table #? – Should this be numbered or is it a continuation of Table 1?

Correct column heading “morfology” to “morphology”.

Table 2 – Treatment Outcome

last column, last page Correct table format in 6th row in right column.

Table 3 – Histology Classification

last row Correct “contains” to “contain”.

Table 4 – Pathology Notes

Heading with “Diagnosis semen analysis” p. 2 6th row from bottom of page Correct “testes” to “testis”.

Table 5 – Cytology Notes

Correct heading from “Ratio’s left side” to “Ratio left side”, and “Ratio’s right side” to “Ratio right side”.

Reviewer #3 (Remarks to the Author):

The authors report on the study of de novo mutations in male infertility in a sample of 185 infertile males.

This is an important contribution to the field due to the large sample of trio cases included. Also, validation was performed in the representative cohort of infertile patients and controls.

Authors specifically address the potential impact of 6 genes, namely RBM5, U2AF2, HNRNPL, CDC5L, CWC27 and TOPAZ1 in infertility.

General comments:

The authors claim that their study provide the first evidence for the role of DNMs in severe male infertility. This conclusion is not grounded since the "paradigm" has been already introduced in the previously published paper which is marginally noted in the manuscript.

The conclusion about the evidence provided is over-interpreted and should be presented more cautiously since most of the rare potentially pathogenic mutations were found in fertile men population as well (some of them even more frequently).

Contrary to intellectual disability the previous and the current study implicate that the DNM paradigm is not responsible for significant "monogenic" causes of male infertility!

Specific comments:

In terms of inclusion criteria It should be stated if testis biopsy was available for all the patients included.

The number of TOPAZ1 heterozygotes/homozygotes in the replication group and fertile population should be stated.

The authors should state specifically if any of the genes has been previously implicated in humans (including GWAS) or animal models.

Rebuttal Letter to Reviewers

We thank the expert reviewers for the feedback on our manuscript and the opportunity to address their comments and question in detail below. Please note that quoted line numbers refer to lines in the final clean version of the manuscript.

Reviewer #1

Major comments:

Comment 1) Abstract. Novelty of the study: A trio-based exome study was published has also been cited, the author cannot neglect it just because of the small sample size. The following statement in the abstract needs to be reworded “Our results provide the first evidence for the role of DNMs in severe male infertility” in the abstract. And in lines 151-153 on page 6, “For replication purposes, unfortunately no other trio-based exome data are available for male infertility, although we note that a pilot study including 13 trios was recently published”, this sentence is conflicted itself and also needs to be reworded. And compare the DNMs even only has 13 trios there.

Reply: Thank you for pointing this out to us. Indeed, we cite a paper from earlier this year in which 11 *de novo* mutations were described in 13 infertile men. That in itself does not provide any evidence for a role of these DNMs in male infertility, an equal number of DNMs would be observed in fertile men. The authors, who also present this work as a pilot study, did not provide any evidence for a role of these mutations in infertility. In contrast, our work in a cohort 14 times larger, provides multiple lines of evidence by comparing the presence and distribution of DNM to those observed in control cohorts, and by providing information about the presence of rare predicted pathogenic mutations in additional cohorts of infertile men. Because of these arguments, we think it is justified to state that this work provides the first convincing evidence for a role of DNM in male infertility. We have added the word ‘convincing’ in the abstract to differentiate this work from the published pilot study. In addition, we discuss the pilot study and its overlap with our work in more detail in lines 180 to 182.

Comment 2) Abstract. New candidate genes: The authors need to be specific on the exact number of new candidate genes been identified, by saying “point to many new candidate genes affecting fertility” in the abstract is not a precise way to present important results and the takeaway for readers.

Reply: We understand this request but don’t think we can be more specific here, the term candidate gene itself is not precisely defined. As we mention on line 53-54 of the abstract, there are 29 genes which have been identified as possibly causative. However, we also list 50 genes classified as ‘Unclear’ and consider these potential candidate genes for male infertility.

Comment 3) Gene intolerance score is only a prediction, which has pitfall of itself (i.e. some known disease genes have low pLI score or even almost 0, see PMID: 30977936). It’s helpful as can provide more support, but not good to use it as direct evidence of pathogenicity. In addition, the authors should also consider including the RVIS score and the LOEUF score.

Reply: We fully agree with the reviewer here and do not claim anywhere in the manuscript that a loss-of-function or missense intolerance score itself is direct evidence of pathogenicity, all of the scores we use serve to support the hypothesis that some of these mutations may cause male

infertility. As requested, we have now added the LOEUF score and obtained the exact same results as for the pLI scores and added information about this analysis (Lines 97-104, lines 328-345 and Supplementary Figure 4a).

We have considered the RVIS score but decided not to include this in our analysis as it is outdated in comparison to both the LOEUF and pLI scores. The latest official release of RVIS scores was back in August 2013 based on data present in ExAC v2. While there were unofficial releases released in 2017, this data was still incomplete as a significant number of genes were still not included. In comparison, pLI and LOEUF scores are based on gnomAD v2.1.1, which included and substantially expanded the ExAC dataset, is regularly updated and provides scores for a much larger percentage of genes.

Comment 4) The authors performed different comparisons for pLI and mis_Z scores but simply say “We observed a significant enrichment of Loss-of-function(LoF) DNMs in LoF-intolerant genes (p-value=1.00x10⁻⁵) as well as predicted pathogenic missense DNMs in missense-intolerant genes (p value=5.01x10⁻⁴)” in one sentence in the abstract. This is misleading and the analyses are problematic. For pLI, the authors compared the median pLI score of genes with LoF DNM in their cohort (n=17) with a public control cohort (n=181), the number of DNMs differ so much in two sets, a direct comparison of the median pLI here could be biased.

The authors need to provide more details on the statistical comparisons applied in the study, like what kind of statistical method was used; and if multiple test correction was applied, the details of how many tests have been corrected should be specified other than just simply say Bonferroni correction was applied, as this could affect the adjusted p-value so as the significance. The authors should provide both the raw and adjusted p-value wherever applied.

Reply: We thank the reviewers for identifying that the statistical tests employed in our analyses had not been sufficiently and clearly described. To address this issue, we have now added additional description and details on the statistical methods used within the Materials and Methods section of the manuscript (Lines 328-345). Briefly, to calculate whether there was a statistically significant enrichment in the DNMs in LoF genes and missense mutations, we applied the methodology described by Lelieveld et al 2016 & Samocha et al. 2014 (Referenced respectively as 64 and 22 in the manuscript). We do this by redistributing the observed number of mutations over a specific set of genes based on their LoF and functional mutation rates, both using pLI and LOEUF scores, and missense mutation rates, using Z-scores. In contrast with the previously mentioned studies, we decided to use a control set of *de novo* mutations to correct for the assumptions that all gnomAD genes are equally likely to be affected by a *de novo* mutation, that some genes are larger, or some are more tolerant to DNMs than others. By applying this proven statistical method, which takes into account the differences in sizes of the gene datasets, and by running 100,000 simulations, we are confident that we can statistically conclude that there was a significant enrichment in LoF DNMs in our patients without performing a direct comparison, which as the reviewers correctly points out would be biased and incorrect.

We also acknowledge that our findings regarding an increase in pathogenic missense DNMs in missense intolerant genes was written in an unclear manner and this has been corrected in the text of the manuscript (lines 187-189 and 345-346). In short, a Mann-Whitney U was used to detect whether the observed pattern of benign missense mutations occurring in more missense-tolerant

genes and pathogenic missense mutations occurring in more missense-intolerant genes, as determined by the gene missense Z-score, was due to chance or statistically correlated.

Comment 5) For *mis_Z*, the authors compared the predicted pathogenic missense DNMs (n=63) with benign ones (n=59) but both from their trios. The significance here is not surprising and doesn't tell much. It's kind of a "cherry-picking" comparison as you are comparing the pathogenic missense to benign ones you have already known. Instead, the comparison between overall missense DNMs in patients and control populations shows no significant difference as in Supplementary Figure 4.

Reply: We agree with the Reviewer that not all the data was presented in a way that allowed readers to reach clear conclusions, our apologies for this. It is correct that the missense Z-score did not differ significantly between all DNM genes in the patient cohort versus the control cohort. This is not unexpected as we expect the majority of our DNM genes not to be linked to infertility, but we have now presented this more clearly in the main text (lines 189-193 and 342-345) and added a supplementary figure (Supplementary Figure 4b).

However, we disagree with the reviewer that 'it is not surprising and doesn't tell much that the missense Z-score for predicted pathogenic DNM genes is significantly higher than that of predicted benign DNM genes' in our patient cohort. This is not cherry-picking, as this analysis was done to identify more clearly whether there is a subgroup of DNM genes that is more likely to be linked to male infertility in our cohort. Importantly, there is no overlap in the way the missense Z-score is calculated for genes and the way the pathogenicity is predicted for DNMs. To further test and show this, we have now added an analysis in which we do the same analysis for the control cohort and do not see a statistical difference between the missense Z-score for genes with predicted pathogenic DNMs and predicted benign DNMs (p-value=0.95, Supplementary Figure 4c).

Comment 6) First of all, the authors need to apply the same filtering and re-annotate the control DNMs the same way as for their cohort to ensure a consistent and unbiased comparison. A different variant could be annotated with different effects by a different method or if on different isoforms. And providing a supplementary table of the variants and the scores used in the analysis is needed.

Reply: To reassure the Reviewer, all variant filtering, in both our patient and control cohorts, was performed in the same way to ensure the consistency required for an unbiased comparison. The Materials and Methods section 'Functional enrichment' has been updated on lines 330-342 to highlight this.

Comment 7) Figure 1 is unnecessarily complicated. The figure presented comparisons that were not mentioned in the paper. Suggest simplifying it to show only the important message. And Figure 2 is too simple to be as an independent figure, suggest combining it with Figure 1 as both are about gene intolerance.

Reply: We thank the Reviewer for this suggestion and have simplified Figure 1 to present the most relevant information. We have also combined the simplified 'Figure 1' and 'Figure 2' to give a more concise overview of our gene intolerance work.

Comment 8) As different exome capture kit and DNA source across the samples, the authors need to provide more details in supplementary on sequencing QC, at least but not limited to the following:

1) A plot of the average target coverage on all samples, indicate in different colors or shapes indicating the capture kit and DNA source (saliva vs blood).

2) Compare the capture regions of the two kits, if there any critical genes or regions in one but not another.

3) Comparison of variant calling on saliva and blood DNA.

Reply: We provide the detailed information requested by the reviewer regarding the quality of the sequencing and the comparability in variant calling. In Supplementary Table 9 and Supplementary Figure 9, now added to the manuscript, we show that the exome kits both cover more than 99% of the known exonic regions.

Differences in the overall target size in each kit are due to the design of the probes, specifically Illumina's Nextera exome kit probes were designed to include larger padding regions surrounding the exons whereas the more recently designed probes in the Twist Bioscience Human exome kit target the exonic regions more efficiently resulting in less padding regions being sequenced. Due to this more efficient design, samples sequenced using the Twist Bioscience kit were sequenced at a higher coverage than those prepared with the Illumina kit. It should be noted that each member of a trio was sequenced using the same exome kit to avoid potential kit performance issues.

Regarding the usage of DNA extracted from 2 different cell types (blood vs saliva), this was a compromise made due to the nature of the recruitment of parents without having them come to the clinics for blood collection. We show in Supplementary Table 10 that there is a small decrease in coverage for saliva samples compared to blood samples. This is due to an increase in the numbers of reads that do not align to the human genome in saliva samples.

Importantly, however, the number of variants called per sample was not statistically different between exome kits or between cell type, allowing reliable DNM calling in all our trios.

Finally, we would like to stress that all DNMs included in this study were independently validated via Sanger sequencing to resolve any uncertainty arising from differences in coverage between probands and parents, sequencing artefacts and variant calling.

Comment 9) The filtering of the candidate mutation on allele balance ($AB > 0.15$) is a bit low, which could introduce somatic mutations and need a careful assessment. The methods (line 39 in "Exome sequencing") mentioned "Variants absent in parental samples", is this mean the parents have no alternative allele ($AB=0$), or just with $AB < 0.15$ that not been classified as variants per the authors' filtering criteria? This needs to be further specified.

Reply: Thank you for bringing this opportunity for clarity to our attention. In the Exome sequencing section in the methods section, we briefly describe the criteria used to call *de novo* mutations in probands (lines 269-275). Most critically, we require for this that variants present in a proband cannot be present at all in the corresponding parental samples (so no single read with the same variant in the parent, $AB=0$). Furthermore, we performed the calling of *de novo* mutations on joint genotyping files containing all probands and parental samples sequenced using the same exome preparation kits and required that recalculated GQ (Phred-scaled probability that the call is correct) for this variant in the proband is at least above 10, and that allelic counts (AC) or allelic frequency

(AF) at this position in the considered population of samples has a maximum of 4 or 0.1%, respectively.

As a cutoff for the variant read percentage we indeed use $< 15\%$, and we do agree with the reviewer that some DNM mutations with a MAF over that % could be somatic mutations. However, in the new Supplementary Table 1, we show that none of our validated DNMs have $MAF < 0.3$, and therefore all DNM described in this work most likely are germline DNMs or arose very early post-zygotically. Indeed, our phasing analysis using long read sequencing did identify 6 DNMs as likely post-zygotic somatic mutations, with 5 of these showing a $MAF < 0.42$ in the exome data, in line with expectations. Larger studies will be required to determine the difference in impact of germline and post-zygotic DNMs on male infertility.

Comment 10) Supplementary Table 1 needs to display in an Excel spreadsheet for easier reuse of the data. The authors also need to provide additional columns for the reference allele and alternative allele, providing the parent-child read depth (DP), allele count (AC), and allele balance (AB) will be also important to increase the future usage of the data.

Reply: We believe that this remark stems from a formatting issue when the paper was transferred to the reviewers but can confirm that supplementary table 1 is indeed displayed as an excel file. We have now included the Minor allele frequency as well as the Read depth for the Proband, Mother and Father. Additionally, all the data necessary for future analyses has been uploaded to the European Genome-Phenome Archive (Study accession number: EGAS00001005417) where the data is available upon request for those who wish to investigate our data further as mentioned in lines 431-434 of the manuscript.

Comment 11) When searching for rare LoF and predicted pathogenic missense mutations in additional infertile and fertile cohorts, the authors should check for all genes with DNM in the study cohort, or at least for the 145 protein-coding DNM genes, but not only the 33 top genes.

Reply: We thank the reviewer for this interesting suggestion. We have extended this work as requested and provide the information now for all 145 genes with coding DNMs (Supplementary Tables 3 to 6). The tables and all relevant statistical analyses have been updated to reflect the increased number of additional patients. Of note, we observed a total of 11 genes with an enrichment of pathogenic missense mutations in infertile men compared to fertile men (Burden Test, $p\text{-value} < 0.05$, Table 2). After correction for multiple testing, however, the only significant enrichment was still observed in the *RBM5* gene. In fact, one additional predicted pathogenic missense variant was found in this gene as additional patients were added to the cohort.

Comment 12) The authors should also look for candidates harboring rare maternally inherited variants, autosomal recessive variants, and X-chromosome hemizygous variants in addition to the DNMs to increase diagnostic yield. Similarly, for CNVs, it is a bit low as there are only 2 de novo CNVs identified. Are there any rare maternally inherited CNVs that harboring any male infertile genes that could explain the patient's phenotype?

Reply: Our study focuses on the role of *de novo* mutations and CNVs in male infertility and some of the additional analyses requested are beyond the scope of this work. As *de novo* mutations mostly

cause dominant disease, we did perform a maternal inheritance analysis to check whether DNM genes were perhaps also affected in other patients by maternally inherited point mutations and copy number variations. We have expanded on this analysis and present our findings regarding maternally inherited SNVs in the text (lines 171-176) and Supplementary Table 2. Unfortunately, no maternally inherited CNVs that affected any of the DNM genes or known male infertility genes were found in our patients. We agree with reviewer that the overall number of *de novo* CNVs identified in this trio cohort is relatively low which we attribute to the limitations of using WES data to investigate CNVs. Whole genome sequencing is currently being performed on this cohort and we hope to obtain a more accurate number of *de novo* CNVs from that work.

Minor comments:

Comment 1) The paper needs to be re-formatted based on the journal style, the language needs to double-check carefully. For example, the gene names should be consistently in italic; gnomAD was wrongly written as “GnomAD”; in methods section “Variant filtration and interpretation” line 64 “Tissue expression was retrieved from the Project The final classification of the genes ...” doesn’t make sense, maybe missed something.

Reply: Thank you for highlighting these issues, we have now addressed this to ensure consistency of abbreviations, italics, and tool names. As part of this, the mentioned text within the methods section has now been rectified in lines 297-304.

Comment 2) Table 2 and supplementary table 2 are very hard to follow, readers could get lost with the many columns. Suggest to reorganize or simplify for easy follow, e.g. the additional cohorts are not necessary to display in individual columns, a sum column will work fine here.

Reply: Thank you for pointing this out. We have now removed the bulk of the data from Table 2 in the main text to allow for easier reading and interpretation. We have only kept the top 11 DNM genes with the highest significance before and after applying the Bonferroni correction to the p-values. We have also removed the separate cohort columns, keeping only the sum column. All the detailed information regarding the findings in additional cohorts can be found in Supplementary Tables 3 to 6.

Comment 3) Variant phasing: The authors need to be clear and specific about the number of DNMs been phased in the 2nd paragraph on page 7.

Reply: The addition of ‘(n=59)’ has been added to the text on line 201. The statistics for the phasing data including number of DNMs phased, the parent of origin and the pathogenicity prediction for the variant, can all be found in Supplementary Table 8.

Comment 4) Some of the references need to be updated to the correct publication (e.g. denovo-db).

Reply: All references have now been checked and the accession date for the *de novo* database has been updated to the date of when data was accessed and retrieved.

Comment 5) The actual rare LoF and missense variants and details worth presenting in a supplementary table, only show counts is not helpful for future data reuse; so as for the DNMs from the control cohort been used in the comparison, and also a good place to provide the underlying intolerance scores.

Reply: The requested information has been added to the supplementary tables 4 and 6.

Reviewer #2

Comment 1) It would add to the contribution of this manuscript were the authors explain the potential clinical utility on how these findings will impact current care of male infertility.

Reply: We thank the reviewer for this suggestion. We have added the following comment on the potential impact of our work at the end of the manuscript, starting on line 223: "Altogether, this will not only help to increase the diagnostic yield for men with infertility but will also enhance our fundamental biological understanding of human reproduction and natural selection. Our results indicate that male infertility will frequently follow a dominant inheritance pattern, and this has impact for disease transmission. Couples that seek treatment for male infertility should be counselled on the risk of transmitting this condition to their offspring, something that is now limited to couples receiving fertility treatment due to Y-chromosome deletions. Male infertility is also increasingly seen as the most visible symptom of a more complex disease with associated comorbidities. Studying the long-term health of men with DNMs in specific genes should help in identifying genotype-phenotype correlations that may impact more than the fertility of these men."

Comment 2) It is difficult to obtain direct genotype-phenotype correlations and establish autosomal dominant pathogenic mechanisms, especially when many genes have only a single variant. Might the authors elaborate on what kind of future work must be carried out in order to solidify the evidence to translate the findings to benefit clinical diagnoses?

Reply: We fully agree that much more work is needed, and the following comment was added to the end of the manuscript, starting on line 217: "In order to achieve this, there is an urgent need to expand on this work as larger studies are essential to identify recurrently DNM genes and further demonstrate the exact contribution of DNMs to male infertility. Modelling studies recently done for developmental disorders showed that more than 350k trios may be required to have ~80% power to detect all haploinsufficient genes causing this disorder. Evidently, these numbers can only be reached by implementing trio-based exome sequencing as a routine diagnostic test and by sharing these diagnostic data with the international research community. This research community will also have the enormous task of functionally validating the impact of these DNMs on spermatogenesis."

Comment 3) It is not clear how the authors arrive at the final classification of the gene variants in the materials and methods. How much weight is put into evidence each from in-silico pathogenicity prediction tools, ACMG guidelines, functional analyses?

Reply: Within the materials and methods section, lines 280-307 describe the methods used to prioritise and finally classify each variant into one of 4 categories. This is based on the likelihood for each specific DNM in each specific gene to cause male infertility. Lines 280-287 describe the initial classification of either 'pathogenic or benign' based on the variant itself and whether the specific base changes are likely to disrupt the gene function. Lines 288-307 then describe how those variants

are further classified using 6 different criteria to investigate the function of the gene itself, its expression locations, mouse models and any connections with known infertility genes.

Comment 4) It should be noted that the small 6kb de novo CNV in Proband_039 is probably not associated with male infertility.

Reply: We agree with the reviewer and this CNV is highlighted in Supplementary Table 1 as ‘Unlikely Causative’.

Comment 5) The authors should explain possible reasons for candidate “rare predicted pathogenic missense variants” or “possibly causative variants” in genes found also in fertile Dutch Men. The authors should discuss the limitations and potential false discovery of causative variants using their approach. For example, SOGA1 is listed in Table 2 as rare potential pathogenic missense mutation but there are also 15 variants in fertile Dutch men. Possibly causative DMN variants in CDK5RAP2 (Table 1) are also found in fertile Dutch men.

Reply: As also pointed out in response to Reviewer 1, comment 3, we use prediction scores and these are not direct evidence of pathogenicity, all of the scores we use serve to support the hypothesis that de novo mutations play a role in male infertility and some of these DNMs may indeed be causing it. As can be seen from updated Supplementary Table 5, indeed many of our DNM genes show the presence of rare predicted pathogenic mutations in the fertile control cohort. Because of that reason, we perform statistical tests to determine whether there is an enrichment of rare predicted pathogenic mutations in infertile vs fertile men. For 11 genes we notice such an enrichment, but after correction for multiple testing RBM5 is the only gene for which we have convincing statistical evidence. For CDK5RAP2, this is clearly not the case and it may very well be that mutations in this gene do not cause male infertility, even though we classified it as possibly causative. For SOGA1, it is important to note that we identified a stop gain *de novo* mutation in our cohort. The presence of predicted pathogenic missense mutations in the fertile control cohort in this gene is not evidence against a LOF mechanism expected for stop mutations, but there are also two fertile men with a LOF mutation. The DNM in this gene was classified as unclear by us. Just to add, it is not unexpected to find rare predicted pathogenic variants in infertility genes in fertile men. It is actually well-known that many healthy individuals carry predicted pathogenic mutations in dominant disease genes, see for example PMID 32272925 or PMID 28471432. While not understood in detail, it can be due to variations in penetrance, something that for infertility could also be influenced by age. All of this will need to be evaluated in larger cohorts. We do mention the need for this at the end of our manuscript.

Comment 6) Was confirmation of biological parents performed for all trios?

Reply: To reassure the Reviewer, yes confirmation of biological parents was performed for all trios both during recruitment as well as following sequencing where relatedness of each sequenced samples was analysed against all other samples. Sequenced samples for which a reported mother or father could not be correctly identified using their sequenced data were excluded from this study and never part of the patient-trio cohort described in this manuscript (we made this more explicit in lines 267-268).

Comment 7) The results of this study are not the first evidence of the role of DNMs in male infertility. Also mentioned by the authors, a pilot study of 13 cases using trio-based exome analysis reported DNMs and candidate genes in idiopathic male infertility was recently published.

Reply: See response to Reviewer 1, comment 1.

Comment 8) As a study of DNMs as a contribution to idiopathic male infertility, it is surprising that 52 men (28%) had a family history of genetic abnormalities or infertility. Did the authors also investigate the landscape of inherited pathogenic variants? What could be possible reasons for the high incidence of family history of genetic abnormalities or infertility?

Reply: We thank the reviewer for these comments, they made us realize that the description and presentation of these data lacked in clarity and precision.

Our study aimed to find the genetic cause in unselected, but mostly sporadic cases of infertility. The goal of these incidence data is to provide insight in a potential inherited (familial) cause of the observed infertility (infertility in the family) and to give additional background information regarding genetic diseases related or unrelated to male infertility (genetic abnormalities in family).

The high percentage of family history noted in these men was reached using the broad criteria that were used for reporting both infertility in the family and genetic abnormalities in family. For both, findings in the second and third-degree family were included. In addition, these data were self-reported and not objectively obtained by a physician. Since infertility is a common disease, our reporting criteria were actually too broad to provide useful insight. We therefore decided to narrow these. For both categories we now limited reporting of findings to the 1st degree family members. In addition, for findings on infertility in the family, we limited these to reports on 1) male infertility and 2) a difficulty of the parents to conceive. Accordingly, we updated the description of these criteria in the Supplementary_Notes_1 (lines 44-46).

Regarding findings related to male infertility in the family, the majority of patients (166/185; 90%) did not report infertility in first degree family members and we consider these sporadic cases of male infertility. None of the patients showed a clear familial pattern of male infertility. However, a total of 19 men (10%) reported these issues within the 1st degree family. Of these, 13 men (7%) reported a brother or father with male factor infertility, and it remains unclear if they share the same aetiology.

Regarding genetic abnormalities in the family, a total of two patients (1%) reported such findings within the 1st degree family. These genetic abnormalities are not related to male infertility and are therefore not relevant for this analysis.

Our study focused on the occurrence of *de novo* mutations related to male infertility in unselected, but mainly sporadic cases. Regarding the landscape of inherited pathogenic variants, an analysis on the recessive causes is currently ongoing. However, this analysis is outside the scope of the current study.

Minor comments:

Comment 1) Table of Mouse models: Use italicized gene symbols throughout. Provide criteria for classification of “Conclusion variant/gene” in table legend.

Reply: We have checked for the correct use of italicized gene symbols throughout. In addition, the classification criteria for the Conclusion variant/gene is highlighted in detail in the materials and methods section, lines 282-310. With this being a large explanation, we feel it is better suited in the materials and methods section.

Comment 2) Table #? Should this be numbered or is it a continuation of Table 1? Correct column heading “morfology” to “morphology”.

Reply: This is indeed a continuation of ‘Supplementary Notes Table 1- Clinical data’, due to formatting for reviewers, it appears that the tables were not shown as an excel sheet which will be their final format. This change to ‘morphology’ has been made.

Comment 3) Table 2 – Treatment Outcome. Last column, last page Correct table format in 6th row in right column.

Reply: Like above, this was likely due to the fact that the Tables provided to the reviewers were not shown as Excel sheet which will be the final format.

Comment 4) Supplementary Figure 7. It is atypical to indicate the father on the right and the mother on the left. Please redraw.

Reply: We thank the reviewer for point out the atypical design of supp. Figure 7 and we have now redrawn this figure to better sit the establish design.

Manuscript

p. 3 line 71 Correct “explaining” to “explain”.

p. 5 line 114 Correct “highlights” to “highlight”.

p. 5 line 128 Insert a comma following RBM5.

Supplementary notes

1.1 Supplementary notes line 29 Delete “with” at end of the line.

1.1 Supplementary notes line 63 Restate “...and the descent of four men was...”

1.1 Supplementary notes line 66 Insert “for” following “patients” and replace “was” with “were.”

1.1 Supplementary notes line 74 Delete the second use of “be”.

1.1 Supplementary notes line 89 Correct “was” to “were”.

1.1 Supplementary notes line 98 Replace “let” with “left”.

1.1 Supplementary notes line 126 Replace “show’ with “shows”.

1.1 Supplementary notes line 134 Add a period at the end of the sentence.

1.1 Supplementary notes line 136 Change “cells” to “cell” and “have” to “is”.

1.1 Supplementary notes line 148 Change “that” to “who”.

Supplementary Figure 2

Correct “atrofy” to “atrophy”, and delete “?” after 50% and after 20%.

Materials and methods:

line 17 Change “was” to “were”.

line 40 Insert a comma after “stringent”.

line 64 Correct partial sentence, “Tissue expression was retrieved from the Project...”.

line 70 Use lower case for sequencing.

line 84 Use lower case for expected.

line 88 Use upper case for Supplementary.

line 100 Is “basecalled” a verb?

line 113 Replace “Variations” with “Variants”.

lines 122 and 129 Use lower case for “g” in gnomAD.

line 130 Replace “Chi-Squared test” with “Chi-square test”.

line 152 Delete comma following “Nagrinaja” and insert a comma following “al.”.

line 164 Change “Consistently” to “Consistent” and insert a comma following “above”.

line 166 Change “coordinated” to “coordinates”.

line 181 Change “is” to “are” and “requires” to “require”.

Supplementary Table 2

Use upper case for Table and italicize gene symbols.

Supplementary Table 3

line 11 Change “on” to “in” prior to both instances of RBM5, delete “the” prior to Proband”, and italicize second instance of RBM5.

Supplementary Table 3 line 12 Italicize RBM5.

Supplementary Table 3 line 13 Change “no” to “not”.

Supplementary Table 5

Use upper case for Table.

Table 1 – Clinical data

Proband_028 Correct “fathers” to “father’s”.

Proband_039 Correct “familie” to “family” and “mothers” to “mother’s”.

Proband_066 Correct “mothers” to ‘mother’s” in two instances.

Proband_129 Correct “mothers” to ‘mother’s”.

Proband_145 Correct “Sisters” to “Sister”.

Proband_161 Correct “irregulair” to “irregular”.

Table 3 – Histology Classification

last row Correct “contains” to “contain”.

Table 4 – Pathology Notes

Heading with “Diagnosis semen analysis” p. 2 6th row from bottom of page Correct “testes” to “testis”.

Table 5 – Cytology Notes

Correct heading from “Ratio’s left side” to “Ratio left side”, and “Ratio’s right side” to “Ratio right side”.

Reply: We thank the reviewer for pointing out these errors to us and we can confirm that all changes have now been made to the manuscript and supporting documents.

Reviewer #3:

Comment 1) The authors claim that their study provide the first evidence for the role of DNMs in severe male infertility. This conclusion is not grounded since the "paradigm" has been already introduced in the previously published paper which is marginally noted in the manuscript.

Reply: See response to Reviewer 1, comment 1.

Comment 2) The conclusion about the evidence provided is over-interpreted and should be presented more cautiously since most of the rare potentially pathogenic mutations were found in fertile men population as well (some of them even more frequently). Contrary to intellectual disability the previous and the current study implicate that the DNM paradigm is not responsible for significant "monogenic" causes of male infertility!

Reply: We respectfully disagree here with the Reviewer. We do not think that we have over-interpreted the data but carefully presented the work, by highlighting the most relevant findings and stringently correcting in our statistical tests for multiple testing. We have adjusted some of our description of our results to indicate to the reader where we found statistical differences and where not, see for example response to Reviewer 1, comment 5. Also, it may not have been clear, but we did not find variation in the fertile control cohort in the exact same position as where we found a rare predicted pathogenic DNM (see lines 290 and 291: Presence of variation in the same position in the fertile male control cohort excludes a DNM from being classified as predicted pathogenic). Indeed, however, it is correct that other rare predicted pathogenic mutations were observed in fertile men in a large number of the genes in which we identified DNMs (see novel Supplementary Tables 4 and 6 for details on mutations identified in fertile and infertile cohorts in these genes). This is not unexpected; it may well be that a number of these DNM genes do not turn out to be genuine male infertility genes. In addition, predicted pathogenic mutations can also occur in healthy individuals in well-known dominant disease genes such as those causing monogenic intellectual disability (see also response to Reviewer 2, comment 5). This is exactly why we have performed the burden test with Bonferroni correction, and we do not over-interpret these data. We highlight *RBM5* as the only gene that is significantly enriched in rare predicted pathogenic variation in infertile men after multiple testing correction. That does not mean that DNMs in many of the other genes do not cause male infertility, but clearly larger cohorts will be needed to evidence this further (as was the case for intellectual disability when we claimed a *de novo* paradigm for this disorder in 2010). Taking all the evidence provided in our manuscript together, we come to the conclusion that *de novo* mutations are likely to play an important role in male infertility. We do mention the importance of expanding on this work at the end of the manuscript, both in expanding the number of trios sequenced as well as the need for functional validation studies.

Specific comments:

Comment 1) In terms of inclusion criteria, it should be stated if testis biopsy was available for all the patients included.

Reply: The Supplementary notes file provides all the necessary information on the patients including breakdowns of age, ethnicity, urological history, infertility phenotype and then all further tests and examinations performed. A detailed breakdown of the 185 patients in relation to potential TESE

operations and the completeness of the samples with regard to histology/cytology sample availability, can be found in 'Supplementary Notes lines 81-86 and 103-110. It is stated that, of all 185 males, 118 had a TESE performed and 86 of these males had histology samples available.

Comment 2) The number of TOPAZ1 heterozygotes/homozygotes in the replication group and fertile population should be stated.

Reply: This information has now been added to the supplementary tables 4 and 6. No rare predicted pathogenic homozygous mutations were found in TOPAZ1 in the replication cohorts or fertile population.

Comment 3) The authors should state specifically if any of the genes has been previously implicated in humans (including GWAS) or animal models.

Reply: The reviewer may have missed this information, but an in-depth analysis was performed looking into gene relations to human fertility and regarding mouse/animal models where infertility has been observed. This information is all located in 'Supplementary table 1' in column 'J' of the spreadsheet. Within the main text, we also highlight the existence of mouse models in a number of our genes of interest for example; *GREB1L* (lines 105-108), *RBM5* (lines 147-153) and *TOPAZ1* (lines 164-166). In the Materials and Methods lines 272 to 288 we also discuss the functional analysis used to categorise variants as either 'Likely Causative', 'Unclear', 'Unlikely Causative' and 'Benign'.

REVIEWERS' COMMENTS

Reviewer #1 (Remarks to the Author):

The authors did a good job to address most of my comments, but there are still several issues not been fully or well addressed, and some new questions arise related to a recent publication. My detailed comments are as follows:

1. Some parts of the abstract are still misleading and oversell for the findings:

1) First, I don't buy the saying of DNMs "explain a significant fraction of the genetic causes of this understudied disorder" even it's hypothesized, there's no strong evidence so far neither from their study itself;

2) The authors still combined two different comparisons in one sentence of "We observed a significant enrichment of Loss-of-function(LoF) DNMs in LoF-intolerant genes (p-value=1.00x10⁻⁵) as well as predicted pathogenic missense DNMs in missense-intolerant genes (p-value=5.01x10⁻⁴)." As commented before, this is misleading as the analysis of LoF variants in cases vs. controls, while it's pathogenic vs. begin for missense variants but all from your cases. As said, this analysis for missense is "cherry-picking" as for sure it will be likely to be significant when you compare pathogenic to begin variants. When people read the abstract, will be easily considered the same significance for both LoF and missense variants, but it's actually not! What's more, the comparison of cases vs. control for missense variants is actually not significant, this is the problem. The author should specify the comparison by adding something like "when compare cases to XX controls" for LoF, and "when compare pathogenic missense variants to the begin ones from cases". Don't try misleading and oversell the findings.

3) Specify and summary what's your findings of RBM5 in this study in this sentence, "One DNM gene identified, RBM5, is an essential regulator of male germ cell pre-mRNA splicing.", and also indicated RBM5 is has been reported as a known gene associated with male infertile.

4) Suggesting adding "with larger sample size" after "... provide the first convincing evidence for the role of DNMs in severe male infertility"; and the ending "point to many new candidate genes affecting fertility" sounds another oversell and don't recommend the use of "many", suggest to soft a bit to something like "point to multiple potential new candidate genes..." if can name few example genes.

2. It's not clear for the original source of the 1,941 control samples retrieved from denovo-db (probably from the SSC autism cohort?). If possible, the authors should try to repeat the intolerance comparisons by difference sex group of controls (e.g. infertile cases vs. male controls, and infertile cases vs. female controls) to see if affect the result as all the cases in this study are males. In addition to the intolerance comparison, it will be also interesting to investigate if your cases harboring more DNMs than the controls.

3. In Table 2: not totally get what's the point of the "Burden test in Fertile Men vs Fertile Women"? or should it be "infertile Men vs Fertile Women"? and how could the adjusted p-values be ranging from 4 to 11? It must be wrongly copied from the column "Total Infertile Men". This is a table about rare missense variants, why show the column of "DNM Classification"? It's clear that only RBM5 reaches marginal corrected significance, why also show other genes? Most genes are with low or even negative mis_Z scores and have a high variant count in the fertile men and women cohort.

4. The title of each section, and of the figures and tables, and the legend needs to be highly improved. For example, the "Results and Discussion" should be just "Results"; "De novo mutation filtering in infertile male trios" suggest to change to something like "Discovery of de novo mutation in infertile male trios"; "Loss-of-function intolerance analysis in de novo mutation genes" change to something like "Intolerance analysis of genes with de novo Loss-of-function mutation"; "Conclusions and implications of our work" should be just "Discussion". Figure 1 is not only about LoF intolerance as the title stated; should also indicate what each bar stands for inside the violin plot; Figure 2: "blue dotted circle" should be "blue dashed circle". Be consistent for syntax, for example, the usage of "DNM genes" in some places reads odd, better to change to "genes with DNM" to be consistent; and "TOPAZ-1 proband" in the title of Figure 3 should be "proband with DNM in TOPAZ1".

5. A recent study (L. Nagirnaja, et al. 2021 NEJM. PMID: 34347949) reported exome sequencing in 924 men with nonobstructive azoospermia, which some of the authors (at least the first and last authors) are also on, and it's obvious that part of this infertile cohort (99 trios?) and the 5,784 Fertile Dutch men also included in Nagirnaja, et al. The authors should comment the sample overlap of those two studies, and indicate the first report of the exome data.

Minor comments:

1. It's better to also indicate the MAF cutoff in brackets for "rare" protein-altering DNMs in the abstract and main when mentioned, even though you already defined in methods.
2. Add "Patients" on Figure 1b as "Patients with Begin Missense DNMs" and "Patients with Begin Missense DNMs"; also add controls on Supplementary Figure 4c "Controls with ...".
3. Add color code on the plot for Supplementary Figure 9.
4. Line 125 specify two of the three tools required to define the "DNM were predicted to be pathogenic".
5. Line 175 should add something of what it suggested, after "of 5,784 fertile men" to close the sentence.
6. Suggest changing the "Burden Test" instead of "Fisher's exact test" to be more specific and also indicate either it's a one-sided or two-sided test.

Reviewer #2 (Remarks to the Author):

This revised manuscript introduces a novel concept of utilizing a trio exome approach to evaluate unexplained male infertility. It focuses on identifying *de novo* pathogenic variants that might be causative of male infertility. A total of 29 genes were identified as possible *de novo* variants that were causative of male infertility. At the same time, the authors present another 50 genes where pathogenicity remains unclear. Based on the findings presented, *de novo* variants suggest a dominant mode of inheritance, which could potentially have an impact in how clinicians counsel patients with male infertility.

This comprehensive article makes clear the urgency for additional research in male infertility. It also makes a strong argument for the need for larger studies to confirm prior findings and to understand genotype-phenotype correlation better.

Overall, the authors did a wonderful job in simplifying the figures and tables. There was also improvement in the grammar throughout the manuscript. In general, the authors fully addressed our comments. This study is relevant as it utilizes an innovative approach to understand male infertility, an often forgotten topic in reproductive genetics.

Reviewer #3 (Remarks to the Author):

In the Introduction section the authors still claim that »Until now, however, a systematic analysis of the role of DNMs in male infertility had not been attempted« and systematically ignore the previously published study in the Andrology. This study tested exactly the same »paradigm«! Such ignorance is not acceptable.

The authors failed to provide a convincing response to the over-interpretation of the study results. Genes with »De novo« mutations identified may be at the moment considered only as putative gene candidates and not an argument for the monogenic cause of male infertility. Similarly than in previously published trio-study the authors do not provide functional studies for their gene candidates. The burden test might be considered as the support for the involvement of these genes in male infertility but does definitely not imply the monogenic cause.

On the other hand, *de novo* mutations are very frequently identified as the monogenic cause of a number of human disorders both affecting reproductive fitness or not.

Consequently, evidence from the current and the first published study on DNMs in male infertility is not sufficient to convincingly claim that the DNM paradigm holds in male infertility. Namely, it looks like that the DNMs are extremely rare in genes identified in both studies (with the potential exception of RBM5).

Similarly, the conclusion that trios should be performed routinely in the clinical setting and infertile couples counselled on the risk for their offspring, is not supported by the current evidence and therefore not acceptable.

In general, the paper brings potentially useful information about new candidate genes for male infertility. However, the authors should refrain from drawing conclusions that are not grounded by the level of evidence provided.

Rebuttal Letter for resubmission Nature Communications manuscript NCOMMS-21-10255B

Reviewer #1

Major comments:

Comment 1) Abstract. First, I don't buy the saying of DNMs "explain a significant fraction of the genetic causes of this understudied disorder" even it's hypothesized, there's no strong evidence so far neither from their study itself;

Reply: Thank you for this suggestion. We have changed the wording to indicate that (1) we limit this hypothesis to severe male infertility (not all of male infertility), and (2) we hypothesize that they may explain a portion of this understudied disorder (instead of a significant fraction).

Comment 2) Abstract. The authors still combined two different comparisons in one sentence of "We observed a significant enrichment of Loss-of-function(LoF) DNMs in LoF-intolerant genes (p -value= 1.00×10^{-5}) as well as predicted pathogenic missense DNMs in missense-intolerant genes (p -value= 5.01×10^{-4})." As commented before, this is misleading as the analysis of LoF variants in cases vs. controls, while it's pathogenic vs. begin for missense variants but all from your cases. As said, this analysis for missense is "cherry-picking" as for sure it will be likely to be significant when you compare pathogenic to begin variants. When people read the abstract, will be easily considered the same significance for both LoF and missense variants, but it's actually not! What's more, the comparison of cases vs. control for missense variants is actually not significant, this is the problem. The author should specify the comparison by adding something like "when compare cases to XX controls" for LoF, and "when compare pathogenic missense variants to the begin ones from cases". Don't try misleading and oversell the findings.

Reply: This part of the abstract has been modified to address this comment as suggested by the Reviewer.

Comment 3) Abstract. Specify and summary what's your findings of RBM5 in this study in this sentence, "One DNM gene identified, RBM5, is an essential regulator of male germ cell pre-mRNA splicing.", and also indicated RBM5 is has been reported as a known gene associated with male infertile.

Reply: We have edited this sentence to the following 'One gene we identify, *RBM5*, is an essential regulator of male germ cell pre-mRNA splicing and has been previously implicated in male infertility in mice.' We have not expanded any further on this gene within this sentence, however the next sentence does then move on to describe the other 6 patients we find who also harbour rare pathogenic missense mutations in this gene. We feel that these two sentences together deliver the message we wish to get across about this gene in a manner that is appropriate for the abstract.

Comment 4) Abstract. Suggesting adding "with larger sample size" after "... provide the first convincing evidence for the role of DNMs in severe male infertility"; and the ending "point to many new candidate genes affecting fertility" sounds another oversell and don't recommend the use of "many", suggest to soft a bit to something like "point to multiple potential new candidate genes..." if can name few example genes.

Reply: We have combined this suggestion with the request from the editor to remove statements on novelty and rephrased as: 'Our results provide evidence for the role of de novo mutations in severe male infertility and point to new candidate genes affecting fertility.'

Comment 5) It's not clear for the original source of the 1,941 control samples retrieved from denovo-db (probably from the SSC autism cohort?). If possible, the authors should try to repeat the intolerance comparisons by difference sex group of controls (e.g. infertile cases vs. male controls, and infertile cases vs. female controls) to see if affect the result as all the cases in this study are males. In addition to the intolerance comparison, it will be also interesting to investigate if your cases harboring more DNMs than the controls.

Reply: We agree with the reviewer that it would be of particular interest to conduct the suggested comparisons, however, the control data retrieved from denovo-db does not contain detailed information for the sex of the controls. The rate of DNMs between cases and controls is difficult to compare as this is impacted by the genomics approaches used and filtering steps applied. For both our cohort and the control cohort this rate is within the normal range of the DNMs expected per individual.

Comment 6) In Table 2: not totally get what's the point of the "Burden test in Fertile Men vs Fertile Women"? or should it be "infertile Men vs Fertile Women"? and how could the adjusted p-values be ranging from 4 to 11? It must be wrongly copied from the column "Total Infertile Men". This is a table about rare missense variants, why show the column of "DNM Classification"? It's clear that only RBM5 reaches marginal corrected significance, why also show other genes? Most genes are with low or even negative mis_Z scores and have a high variant count in the fertile men and women cohort.

Reply: We thank the reviewer for highlighting these points within Table 2. We have now corrected the adjusted p-values to their correct values as there had been an input error. Also, we agree that the burden test in fertile men vs fertile women is not relevant here and removed this from Table 2. However, we believe that the comparison between the fertile men and women is of interest to the readers of this manuscript and as such we have added a short paragraph at the end of the section named "Further analysis in additional cohorts of infertile males ". We have also removed the 'DNM classification' column and the 'Female control data' in Table 2 with similar reasoning, as it does not provide any information required to interpret the data, and full details are still available in Supplementary Data 3 and 5. We did, however, feel it appropriate to show all 11 genes in this Table that were significant before Bonferroni correction. This Reviewer, in the previous round, asked us to expand on the analysis in additional infertile and fertile cohorts by checking for all genes with DNM in the study cohort. Now that we have done this, we of course also have had to correct more stringently for multiple testing. While not all genes displayed here may turn out to be linked to male infertility, we still think that it is promising to see a number of genes (HUWE1, REN, ABLIM1, FUS, CNOT4) for which 4-7 predicted pathogenic mutations are present in the patient cohort but only one or none at all in the control cohort (even though the control cohort is >2 times as large as the patient cohort). This makes the point, as further highlighted in the discussion, that we need to expand on this work as larger studies are essential to robustly identify recurrently DNM genes and further demonstrate the exact contribution of DNMs to male infertility.

Comment 7) The title of each section, and of the figures and tables, and the legend needs to be highly improved. For example, the "Results and Discussion" should be just "Results"; "De novo mutation filtering in infertile male trios" suggest to change to something like "Discovery of de novo mutation in infertile male trios"; "Loss-of-function intolerance analysis in de novo mutation genes"

change to something like “Intolerance analysis of genes with de novo Loss-of-function mutation”; “Conclusions and implications of our work” should be just “Discussion”.

Reply: We have now rectified any issues regarding the formatting of the paper, including the incorrect headings for the results and the discussion. We agree that the subheadings needed significant improvement and so we have taken the reviewers advice to rename the following results subheadings: ‘De novo mutation filtering in infertile male trios’ has become ‘Discovery of de novo mutation in infertile male trios’. ‘Loss-of-function intolerance analysis in de novo mutation genes’ is now ‘Intolerance analysis of genes with de novo Loss-of-function mutation’.

Comment 8) Figure 1 is not only about LoF intolerance as the title stated; should also indicate what each bar stands for inside the violin plot; Figure 2: “blue dotted circle” should be “blue dashed circle”. Be consistent for syntax, for example, the usage of “DNM genes” in some places reads odd, better to change to “genes with DNM” to be consistent; and “TOPAZ-1 proband” in the title of Figure 3 should be “proband with DNM in TOPAZ1”.

Reply: We thank the reviewer for bringing these important details to our attention and we have now addressed them in the respective figure legends.

Comment 9) A recent study (L. Nagirnaja, et al. 2021 NEJM. PMID: 34347949) reported exome sequencing in 924 men with nonobstructive azoospermia, which some of the authors (at least the first and last authors) are also on, and it’s obvious that part of this infertile cohort (99 trios?) and the 5,784 Fertile Dutch men also included in Nagirnaja, et al. The authors should comment the sample overlap of those two studies, and indicate the first report of the exome data.

Reply: It is correct that part of this data was used to investigate the presence of a new candidate recessive male infertility gene PNLDC1, as reported in the NEJM paper. Similarly, we used it to look at the candidate recessive genes M1AP (Wyrwoll et al. 2021 AJHG. PMID 32673564) and PIWIL1 (Oud et al. Cell. 2021. PMID: 33861957). We do not see how this is relevant to comment on in this paper but could add references in the Methods if requested by the Editor.

Minor comments:

Comment 1. It’s better to also indicate the MAF cutoff in brackets for “rare” protein-altering DNMs in the abstract and main when mentioned, even though you already defined in methods.

Reply: As requested the MAF cutoff of <0.1% has been added both into the abstract and the main text.

Comment 2. Add “Patients” on Figure 1b as “Patients with Begin Missense DNMs” and “Patients with Begin Missense DNMs”; also add controls on Supplementary Figure 4c “Controls with ...”.

Reply: The requested classifier has been added to the figure legend of these figures instead of the actual figure to keep with the figure format guidelines.

Comment 3. Add color code on the plot for Supplementary Figure 9.

Reply: The colour code is provided in detail in the figure legend of this figure to prevent cluttering on of the plots which would detract from the data displayed.

Comment 4. Line 125 specify two of the three tools required to define the “DNM were predicted to be pathogenic”.

Reply: The sentence ending 'with 2 of the 3 showing pathogenicity to define a variant as Pathogenic' has been added to make this more clear.

Comment 5. Line 175 should add something of what it suggested, after "of 5,784 fertile men" to close the sentence.

Reply: The sentence ending 'suggesting that these maternally inherited variants are not causative of male infertility' has now been added to clarify the finding.

Comment 6. Suggest changing the "Burden Test" instead of "Fisher's exact test" to be more specific and also indicate either it's a one-sided or two-sided test.

Reply: The suggested change has been made in the main text of the manuscript to make it clear that Fisher's Exact test was used to perform the Burden test in this study.

Reviewer #2 (Remarks to the Author):

No further comments, but we like to thank the Reviewer for his/her kind words and help in improving this manuscript.

Reviewer #3 (Remarks to the Author):

In the Introduction section the authors still claim that »Until now, however, a systematic analysis of the role of DNMs in male infertility had not been attempted« and systematically ignore the previously published study in the Andrology. This study tested exactly the same »paradigm«! Such ignorance is not acceptable.

Reply: We have added a reference to the pilot study in Andrology to the Introduction.

The authors failed to provide a convincing response to the over-interpretation of the study results. Genes with »De novo« mutations identified may be at the moment considered only as putative gene candidates and not an argument for the monogenic cause of male infertility. Similarly than in previously published trio-study the authors do not provide functional studies for their gene candidates. The burden test might be considered as the support for the involvement of these genes in male infertility but does definitely not imply the monogenic cause.

Consequently, evidence from the current and the first published study on DNMs in male infertility is not sufficient to convincingly claim that the DNM paradigm holds in male infertility. Namely, it looks like that the DNMs are extremely rare in genes identified in both studies (with the potential exception of RBM5).

Reply: Supported by comments from this and other Reviewers and the Editor, we have made sure not to over-interpret our results or make claims that cannot be made based on the results presented.

Similarly, the conclusion that trios should be performed routinely in the clinical setting and infertile couples counselled on the risk for their offspring, is not supported by the current evidence and therefore not acceptable.

In general, the paper brings potentially useful information about new candidate genes for male infertility. However, the authors should refrain from drawing conclusions that are not grounded by the level of evidence provided.

Reply: In our resubmission, we were specifically asked by Reviewer 2 to explain the potential clinical utility on how these findings will impact current care of male infertility. We therefore do not think it appropriate to leave this out, but again we have made sure to phrase this in such a way that it is clear that this should happen only after extensive replication in larger cohorts.